# Spectral Methods for Indian Buffet Process Inference

**Hsiao-Yu Fish Tung**
Machine Learning Department
Carnegie Mellon University
Pittsburgh, PA 15213

**Alexander J. Smola**
Machine Learning Department
Carnegie Mellon University and Google
Pittsburgh, PA 15213

## Abstract

The Indian Buffet Process is a versatile statistical tool for modeling distributions over binary matrices. We provide an efficient spectral algorithm as an alternative to costly Variational Bayes and sampling-based algorithms. We derive a novel tensorial characterization of the moments of the Indian Buffet Process proper and for two of its applications. We give a computationally efficient iterative inference algorithm, concentration of measure bounds, and reconstruction guarantees. Our algorithm provides superior accuracy and cheaper computation than comparable Variational Bayesian approach on a number of reference problems.

## 1 Introduction

Inferring the distributions of latent variables is a key tool in statistical modeling. It has a rich history dating back over a century to mixture models for identifying crabs [27] and has served as a key tool for describing diverse sets of distributions ranging from text [10] to images [1] and user behavior [4]. In recent years spectral methods have become a credible alternative to sampling [19] and variational methods [9, 13] for the inference of such structures. In particular, the work of [6, 5, 11, 21, 29] demonstrates that it is possible to infer latent variable structure accurately, despite the problem being nonconvex, thus exhibiting many local minima. A particularly attractive aspect of spectral methods is that they allow for efficient means of inferring the model complexity in the same way as the remaining parameters, simply by thresholding eigenvalue decomposition appropriately. This makes them suitable for nonparametric Bayesian approaches.

While the issue of spectral inference in Dirichlet Distribution is largely settled [6, 7], the domain of nonparametric tools is much richer and it is therefore desirable to see whether the methods can be extended to other models such as the Indian Buffet Process (IBP). This is the main topic of our paper. We provide a full analysis of the tensors arising from the IBP and how spectral algorithms need to be modified, since a degeneracy in the third order tensor requires fourth order terms. To recover the parameters and latent factors, we use Excess Correlation Analysis (ECA) [8] to whiten the higher order tensors and to reduce their dimensionality. Subsequently we employ the power method to obtain symmetric factorization of the higher-order terms. The method provided in this work is simple to implement and has high efficiency in recovering the latent factors and related parameters. We demonstrate how this approach can be used in inferring an IBP structure in the models discussed in [18] and [24]. Moreover, we show that empirically the spectral algorithm provides higher accuracy and lower runtime than variational methods [14]. Statistical guarantees for recovery and stability of the estimates conclude the paper.

**Outline:** Section 2 gives a brief primer on the IBP. Section 3 contains the lower-order moments of IBP and its application on different model. Section 5 discusses concentration of measure of moments. Section 4 applies Excess Correlation Analysis to the moments and it provides the basic structure of this Algorithm. Section 6 shows the empirical performance of our algorithm. Due to space constraints we relegate most derivations and proofs to the appendix.

## 2 The Indian Buffet Process

The Indian Buffet Process defines a distribution over equivalence classes of binary matrices $Z$ with a finite number of rows and a (potentially) infinite number of columns [17, 18]. The idea is that this allows for automatic adjustment of the number of binary entries, corresponding to the number of independent sources, underlying causes, etc. This is a very useful strategy and it has led to many applications including structuring Markov transition matrices [15], learning hidden causes with a bipartite graph [30] and finding latent features in link prediction [26]. $n \in \mathbb{N}$ the number of rows of $Z$, i.e. the number of customers sampling dishes from the " Indian Buffet", let $m_k$ be the number of customers who have sampled dish $k$, let $K_+$ be the total number of dishes sampled, and denote by $K_h$ the number of dishes with a particular selection history $h \in \{0; 1\}^n$. That is, $K_h > 1$ only if there are two or more dishes that have been selected by exactly the same set of customers. Then the probability of generating a particular matrix $Z$ is given by [18]

$$p(Z) = \frac{\alpha^{K_+}}{\prod_h K_h!} \exp\left[-\alpha \sum_{j=1}^{n} \tfrac{1}{j}\right] \prod_{k=1}^{K_+} \frac{(n-m_k)!(m_k-1)!}{n!} \tag{1}$$

Here $\alpha$ is a parameter determining the expected number of nonzero columns in $Z$. Due to the conjugacy of the prior an alternative way of viewing $p(Z)$ is that each column (aka dish) contains nonzero entries $Z_{ij}$ that are drawn from the binomial distribution $Z_{ij} \sim \mathrm{Bin}(\pi_i)$. That is, if we *knew* $K_+$, i.e. if we knew how many nonzero features $Z$ contains, and if we knew the probabilities $\pi_i$, we could draw $Z$ efficiently from it. We take this approach in our analysis: determine $K_+$ and infer the probabilities $\pi_i$ directly from the data. This is more reminiscent of the model used to derive the IBP — a hierarchical Beta-Binomial model, albeit with a variable number of entries:

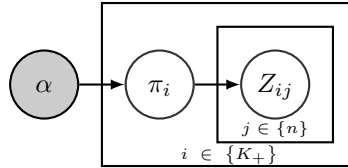

In general, the binary attributes $Z_{ij}$ are *not* observed. Instead, they capture auxiliary structure pertinent to a statistical model of interest. To make matters more concrete, consider the following two models proposed by [18] and [24]. They also serve to showcase the algorithm design in our paper.

**Linear Gaussian Latent Feature Model [18].**   The assumption is that we observe vectorial data $x$. It is generated by linear combination of dictionary atoms $A$ and an associated unknown number of binary causes $z$, all corrupted by some additive noise $\epsilon$. That is, we assume that

$$x = Az + \epsilon \text{ where } \epsilon \sim \mathcal{N}(0, \sigma^2 \mathbf{1}) \text{ and } z \sim \mathrm{IBP}(\alpha). \tag{2}$$

The dictionary matrix $A$ is considered to be fixed but unknown. In this model our goal is to infer both $A$, $\sigma^2$ and the probabilities $\pi_i$ associated with the IBP model. Given that, a maximum-likelihood estimate of $Z$ can be obtained efficiently.

**Infinite Sparse Factor Analysis [24].**   A second model is that of sparse independent component analysis. In a way, it extends (2) by replacing binary attributes with sparse attributes. That is, instead of $z$ we use the entry-wise product $z.*y$. This leads to the model

$$x = A(z.*y) + \epsilon \text{ where } \epsilon \sim \mathcal{N}(0, \sigma^2 \mathbf{1}), z \sim \mathrm{IBP}(\alpha) \text{ and } y_i \sim p(y) \tag{3}$$

Again, the goal is to infer $A$, the probabilities $\pi_i$ and then to associate likely values of $Z_{ij}$ and $Y_{ij}$ with the data. In particular, [24] make a number of alternative assumptions on $p(y)$, namely either that it is iid Gaussian or that it is iid Laplacian. Note that the scale of $y$ itself is not so important since an equivalent model can always be found by rescaling $A$ suitably.

Note that in (3) we used the shorthand $.*$ to denote point-wise multiplication of two vectors in 'Matlab' notation. While (2) and (3) appear rather similar, the latter model is considerably more complex since it not only amounts to a sparse signal but also to an additional multiplicative scale. [24] refer to the model as Infinite Sparse Factor Analysis (isFA) or Infinite Independent Component Analysis (iICA) depending on the choice of $p(y)$ respectively.

# 3 Spectral Characterization

We are now in a position to define the moments of the associated binary matrix. In our approach we assume that $Z \sim \text{IBP}(\alpha)$. We assume that the number of nonzero attributes $k$ is unknown (but fixed). Our analysis begins by deriving moments for the IBP proper. Subsequently we apply this to the two models described above. All proofs are deferred to the Appendix. For notational convenience we denote by $\mathfrak{S}$ the symmetrized version of a tensor where care is taken to ensure that existing multiplicities are satisfied. That is, for a generic third order tensor we set $\mathfrak{S}_6[A]_{ijk} = A_{ijk} + A_{kij} + A_{jki} + A_{jik} + A_{kji} + A_{ikj}$. However, if e.g. $A = B \otimes c$ with $B_{ij} = B_{ji}$, we only need $\mathfrak{S}_3[A]_{ijk} = A_{ijk} + A_{kij} + A_{jki}$ to obtain a symmetric tensor.

## 3.1 Tensorial Moments for the IBP

A degeneracy in the third order tensor requires that we compute a fourth order moment. We can exclude the cases of $\pi_i = 0$ and $\pi_i = 1$ since the former amounts to a nonexistent feature and the latter to a constant offset. We use $M_i$ to denote moments of order $i$ and $S_i$ to denote diagonal(izable) tensors of order $i$. Finally, we use $\pi \in \mathbb{R}^{K+}$ to denote the vector of probabilities $\pi_i$.

**Order 1** This is straightforward, since we have

$$M_1 := \mathbf{E}_z[z] = \pi =: S_1. \tag{4}$$

**Order 2** The second order tensor is given by

$$M_2 := \mathbf{E}_z[z \otimes z] = \pi \otimes \pi + \text{diag}\left(\pi - \pi^2\right) = S_1 \otimes S_1 + \text{diag}\left(\pi - \pi^2\right). \tag{5}$$

Solving for the diagonal tensor we have

$$S_2 := M_2 - S_1 \otimes S_1 = \text{diag}\left(\pi - \pi^2\right). \tag{6}$$

The degeneracies $\{0, 1\}$ of $\pi - \pi^2 = (1 - \pi)\pi$ can be ignored since they amount to non-existent and degenerate probability distributions.

**Order 3** The third order moments yield

$$M_3 := \mathbf{E}_z[z \otimes z \otimes z] = \pi \otimes \pi \otimes \pi + \mathfrak{S}_3\left[\pi \otimes \text{diag}\left(\pi - \pi^2\right)\right] + \text{diag}\left(\pi - 3\pi^2 + 2\pi^3\right) \tag{7}$$

$$= S_1 \otimes S_1 \otimes S_1 + \mathfrak{S}_3\left[S_1 \otimes S_2\right] + \text{diag}\left(\pi - 3\pi^2 + 2\pi^3\right). \tag{8}$$

$$S_3 := M_3 - \mathfrak{S}_3\left[S_1 \otimes S_2\right] + S_1 \otimes S_1 \otimes S_1 = \text{diag}\left(\pi - 3\pi^2 + 2\pi^3\right). \tag{9}$$

Note that the polynomial $\pi - 3\pi^2 + 2\pi^3 = \pi(2\pi - 1)(\pi - 1)$ vanishes for $\pi = \frac{1}{2}$. This is undesirable for the power method — we need to compute a fourth order tensor to exclude this.

**Order 4** The fourth order moments are

$$M_4 := \mathbf{E}_z[z \otimes z \otimes z \otimes z] = S_1 \otimes S_1 \otimes S_1 \otimes S_1 + \mathfrak{S}_6\left[S_2 \otimes S_1 \otimes S_1\right] + \mathfrak{S}_3\left[S_2 \times S_2\right]$$
$$+ \mathfrak{S}_4\left[S_3 \otimes S_1\right] + \text{diag}\left(\pi - 7\pi^2 + 12\pi^3 - 6\pi^4\right)$$
$$S_4 := M_4 - S_1 \otimes S_1 \otimes S_1 \otimes S_1 - \mathfrak{S}_6\left[S_2 \otimes S_1 \otimes S_1\right] - \mathfrak{S}_3\left[S_2 \times S_2\right] + \mathfrak{S}_4\left[S_3 \otimes S_1\right]$$
$$= \text{diag}\left(\pi - 7\pi^2 + 12\pi^3 - 6\pi^4\right). \tag{10}$$

The roots of the polynomial are $\left\{0, \frac{1}{2} - 1/\sqrt{12}, \frac{1}{2} + 1/\sqrt{12}, 1\right\}$. Hence the latent factors and their corresponding $\pi_k$ can be inferred either by $S_3$ or $S_4$.

## 3.2 Application of the IBP

The above derivation showed that if we were able to access $z$ directly, we could infer $\pi$ from it by reading off terms from a diagonal tensor. Unfortunately, this is not quite so easy in practice since $z$ generally acts as a *latent* attribute in a more complex model. In the following we show how the models of (2) and (3) can be converted into spectral form. We need some notation to indicate multiplications of a tensor $M$ of order $k$ by a set of matrices $A_i$.

$$[T(M, A_1, \ldots, A_k)]_{i_1, \ldots i_k} := \sum_{j_1, \ldots j_k} M_{j_1, \ldots j_k}[A_1]_{i_1 j_1} \cdot \ldots \cdot [A_k]_{i_k j_k}. \tag{11}$$

Note that this includes matrix multiplication. For instance, $A_1^\top M A_2 = T(M, A_1, A_2)$. Also note that in the special case where the matrices $A_i$ are vectors, this amounts to a reduction to a scalar. Any such reduced dimensions are assumed to be dropped implicitly. The latter will become useful in the context of the tensor power method in [6].

**Linear Gaussian Latent Factor Model.** When dealing with (2) our goal is to infer both $A$ and $\pi$. The main difference is that rather than observing $z$ we have $Az$, hence all tensors are colored. Moreover, we also need to deal with the terms arising from the additive noise $\epsilon$. This yields

$$S_1 := M_1 = T(\pi, A) \tag{12}$$

$$S_2 := M_2 - S_1 \otimes S_1 - \sigma^2 \mathbf{1} = T(\mathrm{diag}(\pi - \pi^2), A, A) \tag{13}$$

$$S_3 := M_3 - S_1 \otimes S_1 \otimes S_1 - \mathfrak{S}_3 [S_1 \otimes S_2] - \mathfrak{S}_3 [m_1 \otimes \mathbf{1}] \tag{14}$$
$$= T\left(\mathrm{diag}\left(\pi - 3\pi^2 + 2\pi^3\right), A, A, A\right)$$

$$S_4 := M_4 - S_1 \otimes S_1 \otimes S_1 \otimes S_1 - \mathfrak{S}_6 [S_2 \otimes S_1 \otimes S_1] - \mathfrak{S}_3 [S_2 \otimes S_2] - \mathfrak{S}_4 [S_3 \otimes S_1] \tag{15}$$
$$- \sigma^2 \mathfrak{S}_6 [S_2 \otimes \mathbf{1}] - m_4 \mathfrak{S}_3 [\mathbf{1} \otimes \mathbf{1}]$$
$$= T\left(\mathrm{diag}\left(-6\pi^4 + 12\pi^3 - 7\pi^2 + \pi\right), A, A, A, A\right)$$

Here we used the auxiliary statistics $m_1$ and $m_4$. Denote by $v$ the eigenvector with the smallest eigenvalue of the covariance matrix of $x$. Then the auxiliary variables are defined as

$$m_1 := \mathbf{E}_x\left[x \left\langle v, (x - \mathbf{E}[x])\right\rangle^2\right] \qquad = \sigma^2 T(\pi, A) \tag{16}$$

$$m_4 := \mathbf{E}_x\left[\left\langle v, (x - \mathbf{E}_x[x])\right\rangle^4\right]/3 \qquad = \sigma^4. \tag{17}$$

These terms are used in a tensor power method to infer both $A$ and $\pi$ (Appendix A has a derivation).

**Infinite Sparse Factor Analysis.** Using the model of (3) it follows that $z$ is a *symmetric* distribution with mean 0 provided that $p(y)$ has this property. From that it follows that the first and third order moments and tensors vanish, i.e. $S_1 = 0$ and $S_3 = 0$. We have the following statistics:

$$S_2 := M_2 - \sigma^2 \mathbf{1} = T(c \cdot \mathrm{diag}(\pi), A, A) \tag{18}$$

$$S_4 := M_4 - \mathfrak{S}_3 [S_2 \otimes S_2] - \sigma^2 \mathfrak{S}_6 [S_2 \otimes \mathbf{1}] - m_4 \mathfrak{S}_3 [\mathbf{1} \otimes \mathbf{1}] = T\left(\mathrm{diag}(f(\pi)), A, A, A, A\right). \tag{19}$$

Here $m_4$ is defined as in (17). Whenever $p(y)$ in (3) is Gaussian, we have $c = 1$ and $f(\pi) = \pi - \pi^2$. Moreover, whenever $p(y)$ follows the Laplace distribution, we have $c = 2$ and $f(\pi) = 24\pi - 12\pi^2$.

**Lemma 1** *Any linear model of the form (2) or (3) with the property that $\epsilon$ is symmetric and satisfies $\mathbf{E}[\epsilon^2] = \mathbf{E}\left[\epsilon_{\mathrm{Gauss}}^2\right]$ and $\mathbf{E}[\epsilon^4] = \mathbf{E}\left[\epsilon_{\mathrm{Gauss}}^4\right]$ the same properties for $y$, will yield the same moments.*

**Proof** This follows directly from the fact that $z$, $\epsilon$ and $y$ are independent and that the latter two have zero mean and are symmetric. Hence the expectations carry through regardless of the actual underlying distribution. ∎

## 4 Parameter Inference

Having derived symmetric tensors that contain both $A$ and polynomials of $\pi$, we need to separate those two factors and the additive noise, as appropriate. In a nutshell the approach is as follows: we first identify the noise floor using the assumption that the number of nonzero probabilities in $\pi$ is lower than the dimensionality of the data. Secondly, we use the noise-corrected second order tensor to whiten the data. This is akin to methods used in ICA [12]. Finally, we perform power iterations on the data to obtain $S_3$ and $S_4$, or rather, their applications to data. Note that the eigenvalues in the re-scaled tensors differ slightly since we use $S_2^{\dagger \frac{1}{2}} x$ directly rather than $x$.

**Robust Tensor Power Method** Our reasoning follows that of [6]. It is our goal to obtain an *orthogonal* decomposition of the tensors $S_i$ into an orthogonal matrix $V$ together with a set of corresponding eigenvalues $\lambda$ such that $S_i = T[\mathrm{diag}(\lambda), V^\top, \ldots, V^\top]$. This is accomplished by generalizing the Rayleigh quotient and power iterations as described in [6, Algorithm 1]:

$$\theta \leftarrow T[S, \mathbf{1}, \theta, \ldots, \theta] \text{ and } \theta \leftarrow \|\theta\|^{-1} \theta. \tag{20}$$

---

**Algorithm 1** Excess Correlation Analysis for Linear-Gaussian model with IBP prior

---
**Inputs:** the moments $M_1, M_2, M_3, M_4$.

 1: **Infer K and $\sigma^2$:**
 2: Optionally find a subspace $R \in \mathbb{R}^{d \times K'}$ with $K < K'$ by random projection.

$$\text{Range}\,(R) = \text{Range}\,(M_2 - M_1 \otimes M_1) \text{ and project down to } R$$

 3: Set $\sigma^2 := \lambda_{\min}\,(M_2 - M_1 \otimes M_1)$
 4: Set $S_2 = \left(M_2 - M_1 \otimes M_1 - \sigma^2 \mathbf{1}\right)_\epsilon$ by truncating to eigenvalues larger than $\epsilon$
 5: Set $K = \text{rank}\, S_2$
 6: Set $W = U\Sigma^{-\frac{1}{2}}$, where $[U, \Sigma] = \text{svd}(S_2)$
 7: **Whitening:** (best carried out by preprocessing $x$)
 8: Set $W_3 := T(S_3, W, W, W)$
 9: Set $W_4 := T(S_4, W, W, W, W)$
10: **Tensor Power Method:**
11: Compute generalized eigenvalues and vectors of $W_3$.
12: Keep all $K_1 \le K$ (eigenvalue, eigenvector) pairs $(\lambda_i, v_i)$ of $W_3$
13: Deflate $W_4$ with $(\lambda_i, v_i)$ for all $i \le K_1$
14: Keep all $K - K_1$ (eigenvalue, eigenvector) pairs $(\lambda_i, v_i)$ of deflated $W_4$
15: **Reconstruction:** With corresponding eigenvalues $\{\lambda_1, \cdots, \lambda_K\}$, return the set $A$:

$$A = \left\{ \frac{1}{Z_i} \left(W^\dagger\right)^\top v_i : v_i \in \Lambda \right\} \tag{21}$$

where $Z_i = \sqrt{\pi_i - \pi_i^2}$ with $\pi_i = f^{-1}(\lambda_i)$. $f(\pi) = \frac{-2\pi + 1}{\sqrt{\pi - \pi^2}}$ if $i \in [K_1]$ and $f(\pi) = \frac{6\pi^2 - 6\pi + 1}{\pi - \pi^2}$ otherwise. (The proof of Equation (21) is provided in the Appendix.)

---

In a nutshell, we use a suitable number of random initialization $l$, perform a few iterations ($v$) and then proceed with the most promising candidate for another $d$ iterations. The rationale for picking the best among $l$ candidates is that we need a high probability guarantee that the selected initialization is non-degenerate. After finding a good candidate and normalizing its length we deflate (i.e. subtract) the term from the tensor $S$.

**Excess Correlation Analysis (ECA)** The algorithm for recovering $A$ is shown in Algorithm 1. We first present the method of inferring the number of latent features, $K$, which can be viewed as the rank of the covariance matrix. An efficient way of avoiding eigendecomposition on a $d \times d$ matrix is to find a low-rank approximation $R \in \mathbb{R}^{d \times K'}$ such that $K < K' \ll d$ and $R$ spans the same space as the covariance matrix. One efficient way to find such matrix is to set $R$ to be

$$R = (M_2 - M_1 \times M_1)\,\Theta, \tag{22}$$

where $\Theta \in \mathbb{R}^{d \times K'}$ is a random matrix with entries sampled independently from a standard normal. This is described, e.g. by [20]. Since there is noise in the data, it is not possible that we get exactly $K$ non-zero eigenvalues with the remainder being constant at noise floor $\sigma^2$. An alternative strategy to thresholding by $\sigma^2$ is to determine $K$ by seeking the largest slope on the curve of sorted eigenvalues.

Next, we whiten the observations by multiplying data with $W \in \mathrm{R}^{d \times K}$. This is computationally efficient, since we can apply this directly to $x$, thus yielding third and fourth order tensors $W_3$ and $W_4$ of size $k$. Moreover, approximately factorizing $S_2$ is a consequence of the decomposition and random projection techniques arising from [20].

To find the singular vectors of $W_3$ and $W_4$ we use the robust tensor power method, as described above. From the eigenvectors we found in the last step, $A$ could be recovered with Equation 21. The fact that this algorithm only needs projected tensors makes it very efficient. Streaming variants of the robust tensor power method are subject of future research.

**Further Details on the projected tensor power method.** Explicitly calculating tensors $M_2, M_3, M_4$ is not practical in high dimensional data. It may not even be desirable to compute the projected variants of $M_3$ and $M_4$, that is, $W_3$ and $W_4$ (after suitable shifts). Instead, we can use

the analog of a kernel trick to simplify the tensor power iterations to

$$W^\top T(M_l, \mathbf{1}, Wu, \ldots, Wu) = \frac{1}{m} \sum_{i=1}^{m} W^\top x_i \langle x_i, Wu \rangle^{l-1} = \frac{W^\top}{m} \sum_{i=1}^{m} x_i \langle W^\top x_i, u \rangle^{l-1}$$

By using incomplete expansions memory complexity and storage are reduced to $O(d)$ per term. Moreover, precomputation is $O(d^2)$ and it can be accomplished in the first pass through the data.

## 5 Concentration of Measure Bounds

There exist a number of concentration of measure inequalities for *specific* statistical models using rather specific moments [8]. In the following we derive a general tool for bounding such quantities, both for the case where the statistics are bounded and for unbounded quantities alike. Our analysis borrows from [3] for the bounded case, and from the average-median theorem, see e.g. [2].

### 5.1 Bounded Moments

We begin with the analysis for bounded moments. Denote by $\phi : \mathcal{X} \to \mathcal{F}$ a set of statistics on $\mathcal{X}$ and let $\phi_l$ be the $l$-times tensorial moments obtained from $l$.

$$\phi_1(x) := \phi(x); \quad \phi_2(x) := \phi(x) \otimes \phi(x); \quad \phi_l(x) := \phi(x) \otimes \ldots \otimes \phi(x) \tag{23}$$

In this case we can define inner products via

$$k_l(x, x') := \langle \phi_l(x), \phi_l(x') \rangle = T[\phi_l(x), \phi(x'), \ldots, \phi(x')] = \langle \phi(x), \phi(x') \rangle^l = k^l(x, x')$$

as reductions of the statistics of order $l$ for a kernel $k(x, x') := \langle \phi(x), \phi(x') \rangle$. Finally, denote by

$$M_l := \mathbf{E}_{x \sim p(x)}[\phi_l(x)] \text{ and } \hat{M}_l := \frac{1}{m} \sum_{j=1}^{m} \phi_l(x_j) \tag{24}$$

the expectation and empirical averages of $\phi_l$. Note that these terms are identical to the statistics used in [16] whenever a polynomial kernel is used. It is therefore not surprising that an analogous concentration of measure inequality to the one proven by [3] holds:

**Theorem 2** *Assume that the sufficient statistics are bounded via $\|\phi(x)\| \le R$ for all $x \in \mathcal{X}$. With probability at most $1 - \delta$ the following guarantee holds:*

$$\Pr \left\{ \sup_{u:\|u\| \le 1} \left| T(M_l, u, \cdots, u) - T(\hat{M}_l, u, \cdots, u) \right| > \epsilon_l \right\} \le \delta \text{ where } \epsilon_l \le \frac{\left[ 2 + \sqrt{-2 \log \delta} \right] R^l}{\sqrt{m}}.$$

Using Lemma 1 this means that we have concentration of measure immediately for the moments $S_1, \ldots S_4$.Details are provided in the appendix. In particular, we need a chaining result (Lemma 4) that allows us to compute bounds for products of terms efficiently. By utilizing an approach similar to [8], overall guarantees for reconstruction accuracy can be derived.

### 5.2 Unbounded Moments

We are interested in proving concentration of the following four tensors in (13), (14), (15) and one scalar in (27). Whenever the statistics are unbounded, concentration of moment bounds are less trivial and require the use of subgaussian and gaussian inequalities [22]. We derive a bound for fourth-order subgaussian random variables (previous work only derived up to third order bounds). Lemma 5 and 6 has details on how to obtain such guarantees. We further get the bounds for the tensors based on the concentration of moment in Lemma 7 and 8. Bounds for reconstruction accuracy of our algorithm are provided. The full proof is in the Appendix.

**Theorem 3** *(Reconstruction Accuracy) Let $\varsigma_k[S_2]$ be the $k-th$ largest singular value of $S_2$. Define $\pi_{min} = \operatorname{argmax}_{i \in [K]} |\pi_i - 0.5|$, $\pi_{max} = \operatorname{argmax}_{i \in [K]} \pi_i$ and $\tilde{\pi} = \prod_{\{i:\pi_i \le 0.5\}} \pi_i \prod_{\{i:\pi_i > 0.5\}} (1 -$*

$\pi_i$). *Pick any* $\delta, \epsilon \in (0,1)$. *There exists a polynomial* $\mathrm{poly}(\cdot)$ *such that if sample size* $m$ *statisfies*

$$m \geq \mathrm{poly}\left(d, K, \frac{1}{\epsilon}, \log(1/\delta), \frac{1}{\tilde{\pi}}, \frac{\varsigma_1[S_2]}{\varsigma_K[S_2]}, \frac{\sum\limits_{i=1}^{K}\|A_i\|_2^2}{\varsigma_K[S_2]}, \frac{\sigma^2}{\varsigma_K[S_2]}, \frac{1}{\sqrt{\pi_{min} - \pi_{min}^2}}, \frac{\pi_{max}}{\sqrt{\pi_{max} - \pi_{max}^2}}\right)$$

*with probability greater than* $1 - \delta$, *there is a permutation* $\tau$ *on* $[K]$ *such that the* $\hat{A}$ *returns by Algorithm 1 satifies* $\left\|\hat{A}_{\tau(i)} - A_i\right\| \leq \left(\|A_i\|_2 + \sqrt{\varsigma_1[S_2]}\right)\epsilon$ *for all* $i \in [K]$.

## 6 Experiments

We evaluate the algorithm on a number of problems suitable for the two models of (2) and (3). The problems are largely identical to those put forward in [18] in order to keep our results comparable with a more traditional inference approach. We demonstrate that our algorithm is faster, simpler, and achieves comparable or superior accuracy.

**Synthetic data**    Our goal is to demonstrate the ability to recover latent structure of generated data. Following [18] we generate images via linear noisy combinations of $6 \times 6$ templates. That is, we use the binary additive model of (2). The goal is to recover both the above images and to assess their respective presence in observed data. Using an additive noise variance of $\sigma^2 = 0.5$ we are able to recover the original signal quite accurately (from left to right: true signal, signal inferred from 100 samples, signal inferred from 500 samples). Furthermore, as the second row indicates, our algorithm also correctly infers the attributes present in the images.

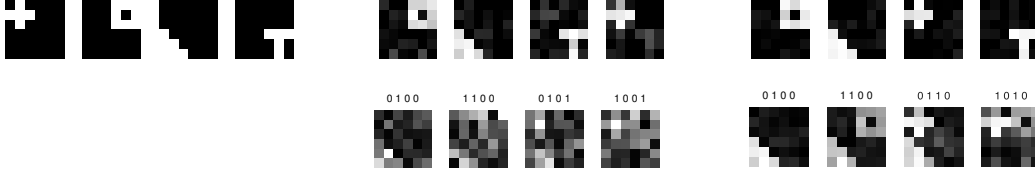

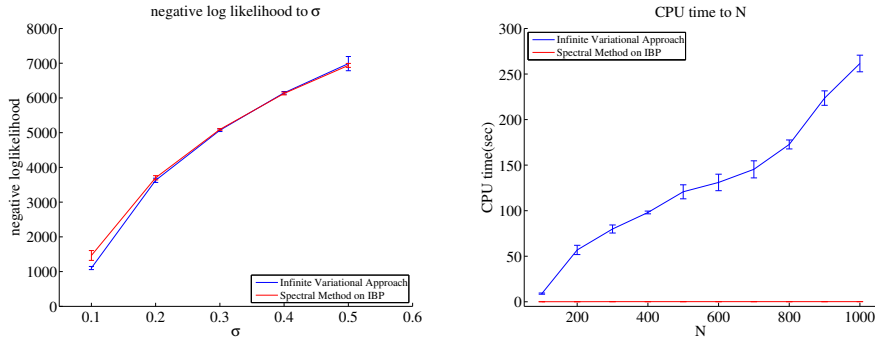

For a more quantitative evaluation we compared our results to the infinite variational algorithm of [14]. The data is generated using $\sigma \in \{0.1, 0.2, 0.3, 0.4, 0.5\}$ and with sample size $n \in \{100, 200, 300, 400, 500\}$. Figure 1 shows that our algorithm is faster and comparatively accurate.

Figure 1: Comparison to infinite variational approach. The first plot compares the test negative log likelihood training on $N = 500$ samples with different $\sigma$. The second plot shows the CPU time to data size, $N$, between the two methods.

**Image Source Recovery**    We repeated the same test using 100 photos from [18]. We first reduce dimensionality on the data set by representing the images with 100 principal components and apply our algorithm on the 100-dimensional dataset (see Algorithm 1 for details). Figure 2 shows the result. We used 10 initial iterations 50 random seeds and 30 final iterations 50 in the Robust Power Tensor Method. The total runtime was 0.2788s.

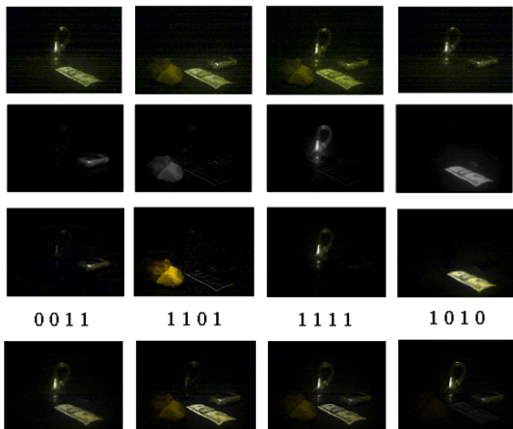

0011    1101    1111    1010

Figure 2: Results of modeling 100 images from [18] of size $240 \times 320$ by model (2). Row 1: four sample images containing up to four objects ($20 bill, Klein bottle, prehistoric handaxe, cellular phone). An object basically appears in the same location, but some small variation noise is generated because the items are put into scene by hand; Row 2: Independent attributes, as determined by infinite variational inference of [14] (note, the results in [18] are black and white only); Row 3: Independent attributes, as determined by spectral IBP; Row 4: Reconstruction of the images via spectral IBP. The binary superscripts indicate the items identified in the image.

Original G     Spectral isFA     MCMC

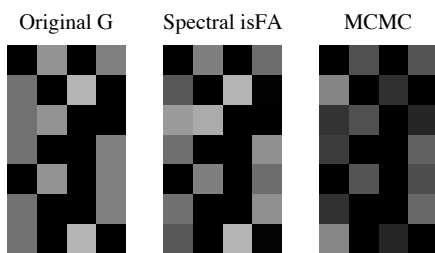

Figure 3: Recovery of the source matrix $A$ in model (3) when comparing MCMC sampling and spectral methods. MCMC sampling required 1.72 seconds and yielded a Frobenius distance $\|A - A_{\mathrm{MCM}}\|_F = 0.77$. Our spectral algorithm required 0.77 seconds to achieve a distance $\|A - A_{\mathrm{Spectral}}\|_F = 0.31$.

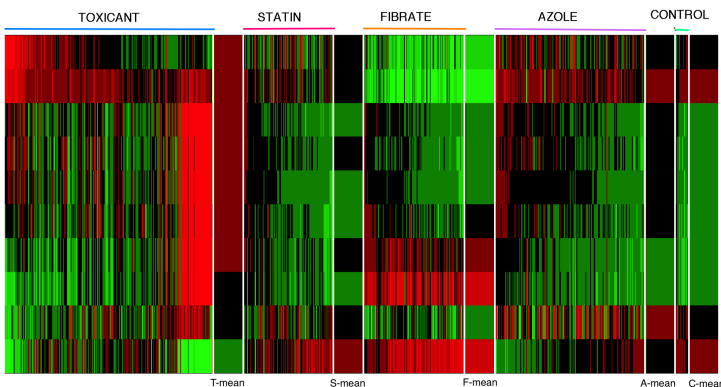

Figure 4: Gene signatures derived by the spectral IBP. They show that there are common hidden causes in the observed expression levels, thus offering a considerably simplified representation.

**Gene Expression Data**    As a first sanity check of the feasibility of our model for (3), we generated synthetic data using $x \in \mathbb{R}^7$ with $k = 4$ sources and $n = 500$ samples, as shown in Figure 3.

For a more realistic analysis we used a microarray dataset. The data consisted of 587 mouse liver samples detecting 8565 gene probes, available as dataset GSE2187 as part of NCBI's Gene Expression Omnibus www.ncbi.nlm.nih.gov/geo. There are four main types of treatments, including Toxicant, Statin, Fibrate and Azole. Figure 4 shows the inferred latent factors arising from expression levels of samples on 10 derived gene signatures. According to the result, the group of fibrate-induced samples and a small group of toxicant-induced samples can be classified accurately by the special patterns. Azole-induced samples have strong positive signals on gene signatures 4 and 8, while statin-induced samples have strong positive signals only on the 9 gene signatures.

**Summary**    In this paper we introduced a spectral approach to inferring latent parameters in the Indian Buffet Process. We derived tensorial moments for a number of models, provided an efficient inference algorithm, concentration of measure theorems and reconstruction guarantees. All this is backed up by experiments comparing spectral and MCMC methods.

We believe that this is a first step towards expanding spectral nonparametric tools beyond the more common Dirichlet Process representations. Applications to more sophisticated models, larger datasets and efficient implementations are subject for future work.

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
