[Supplementary Material]

# A Derivation of Moments for the IBP (Section 3)

## A.1 Moments for the Linear Gaussian Latent Feature Model

Key to the derivation is the fact that $z$ and $\epsilon$ are independent random variables, hence their expectations can be taken independently.

**Order 1 tensor:** By using Equation (4), we have

$$S_1 := M_1 = \mathbf{E}_x\left[x\right] = \mathbf{E}_z\left[Az + \epsilon\right] = A\mathbf{E}_z\left[z\right] = T(\pi, A). \tag{25}$$

To infer the *number* of latent variables $k$ and deal with the noise term, we need to determine the rank of the covariance matrix $\mathbf{E}_x\left[(x - \mathbf{E}_x[x]) \otimes (x - \mathbf{E}_x[x])\right]$. Because there is additive noise, the smallest $(d - K)$ eigenvalues will not be exactly zero. Instead, they amount to the variance arising from $\epsilon$ since

$$\mathrm{cov}[Az + \epsilon] = A^\top \mathrm{cov}[z]A + \mathrm{cov}[\epsilon]. \tag{26}$$

Consequently the smallest eigenvalues of the covariance matrix of $x$ allow us to read off the variance $\sigma^2$: for any normal vector $v$ corresponding to the $d - k$ smallest eigenvalues we have

$$\mathbf{E}_x\left[\left(v^\top\left(x - \mathbf{E}\left[X\right]\right)\right)^2\right] = v^\top A^\top \mathrm{cov}[z]Av + v^\top \mathrm{cov}[\epsilon]v = \sigma^2. \tag{27}$$

**Order 2 tensor:** For the second-order tensor, we plug in (6) and use independence of $z$ and $\epsilon$. Linear terms in $\epsilon$ vanish. Hence we get

$$\begin{aligned} M_2 &= \mathbf{E}_x\left[x \otimes x\right] = T\left(\mathbf{E}_z\left[z \otimes z\right], A, A\right) + \sigma^2 \mathbf{1} = T\left(\pi \otimes \pi + \mathrm{diag}\left(\pi - \pi^2\right), A, A\right) + \sigma^2 \mathbf{1} \\ &= S_1 \otimes S_1 + T\left(\mathrm{diag}\left(\pi - \pi^2\right), A, A\right) + \sigma^2 \mathbf{1}. \end{aligned} \tag{28}$$

This yields the statement in Equation (13).

**Order 3 tensor:** As before, denote by $v$ an eigenvector corresponding to the $(d - k)$ smallest eigenvalues, i.e. $v^\top A = 0$. We first define an auxiliary term

$$\begin{aligned} m_1 &:= \mathbf{E}_x\left[x\left(v^\top\left(x - \mathbf{E}[x]\right)\right)^2\right] = \mathbf{E}_x\left[x\left(v^\top\left(A\left(z - \pi\right) + \varepsilon\right)\right)^2\right] \\ &= \mathbf{E}_x\left[x\left(v^\top \varepsilon\right)^2\right] = \sigma^2 T(\pi, A). \end{aligned} \tag{29}$$

Since the Normal Distribution is symmetric, only even moments of $\epsilon$ survive. Using (9), the third order moments yield

$$M_3 = \mathbf{E}_x\left[x \otimes x \otimes x\right] = \mathbf{E}_z\left[Az \otimes Az \otimes Az\right] + \mathbf{E}_z\left[\mathfrak{S}_3\left[Az \otimes \epsilon \otimes \epsilon\right]\right] \tag{30}$$

$$= T\left(\mathbf{E}_z[z \otimes z \otimes z], A, A, A\right) + \mathfrak{S}_3\left(m_1 \otimes \mathbf{1}\right) \tag{31}$$

$$= S_1 \otimes S_1 \otimes S_1 + \mathfrak{S}_3\left[S_1 \otimes S_2\right] + T\left(\mathrm{diag}\left(\pi - 3\pi_i^2 + 2\pi_i^3\right), A, A, A\right) + \mathfrak{S}_3\left(m_1 \otimes \mathbf{1}\right)$$

Thus, we get Equation (14).

**Order 4 tensor:** We obtain the fourth-order tensor by first calculating an auxiliary variable related to the additive noise term

$$m_4 := \mathbf{E}_x\left[\left(v^\top\left(x - \mathbf{E}_x\left[x\right]\right)\right)^4\right]/3 = \mathbf{E}\left[\left(v^\top \epsilon\right)^4\right]/3 = \sigma^4. \tag{32}$$

Here the last equality followed from the isotropy of Gaussians. With Equation (10), the forth order moments are

$$\begin{aligned} M_4 &= \mathbf{E}_x\left[x \otimes x \otimes x \otimes x\right] \\ &= \mathbf{E}_z\left[Az \otimes Az \otimes Az \otimes Az\right] + \mathbf{E}_z\left[\mathfrak{S}_6\left[Az \otimes Az \otimes \epsilon \otimes \epsilon\right]\right] + \mathbf{E}[\epsilon \otimes \epsilon \otimes \epsilon \otimes \epsilon] \\ &= T\left(\mathbf{E}_z\left[z \otimes z \otimes z \otimes z\right], A, A, A, A\right) + \sigma^2 \mathfrak{S}_6\left[S_2 \otimes \mathbf{1}\right] + \sigma^4 \mathfrak{S}_3\left[\mathbf{1} \otimes \mathbf{1}\right] \\ &= S_1 \otimes S_1 \otimes S_1 \otimes S_1 + \mathfrak{S}_6\left[S_2 \otimes S_1 \otimes S_1\right] + \mathfrak{S}_3\left[S_2 \times S_2\right] + \mathfrak{S}_4\left[S_3 \otimes S_1\right] \\ &\quad + T\left(\mathrm{diag}\left(-6\pi^4 + 12\pi^3 - 7\pi^2 + \pi\right), A, A, A\right) + \sigma^2 \mathfrak{S}_6\left[S_2 \otimes \mathbf{1}\right] + m_4 \mathfrak{S}_3\left[\mathbf{1} \otimes \mathbf{1}\right]. \end{aligned}$$

$$\tag{33}$$
$$\tag{34}$$
$$\tag{35}$$

## A.2 Moments for the Infinite Sparse Factor Analysis Model

Since both $Y$ and $\epsilon$ are symmetric and have zero mean, the odd order tensors vanish. That is $M_1 = 0$ and $M_3 = 0$. It suffices for us to focus on the even terms.

**Order 2 tensor:** Using covariance matrix of (6) yields

$$M_2 = \mathbf{E}_x\left[x \otimes x\right] = T\left(\mathbf{E}_z[(z \odot y) \otimes (z \odot y)], G, G\right) + \sigma^2 \mathbf{1} \tag{36}$$

$$= T\left(\left(\mathbf{E}_z[z \otimes z] \odot \mathbf{E}_y\left[y^2\right]\mathbf{1}\right), G, G\right) + \sigma^2 \mathbf{1} \tag{37}$$

$$= T\left(\left(\pi \otimes \pi + \mathrm{diag}\left(\pi - \pi^2\right)\right) \odot \mathbf{E}_y\left[y^2\right]\mathbf{1}, G, G\right) + \sigma^2 \mathbf{1} \tag{38}$$

$$= T\left(\mathbf{E}_y\left[y^2\right]\mathrm{diag}\left(\pi\right), G, G\right) + \sigma^2 \mathbf{1} = T\left(\mathrm{diag}\left(\pi\right), G, G\right) + \sigma^2 \mathbf{1}, \tag{39}$$

As before, the variance $\sigma^2$ of $\epsilon$ can be inferred by Equation (27). Here we get Equation (18).

**Order 4 tensor:** With Equation (10) and $\mathbf{E}_y\left[y^4\right] = 3$, we have

$$M_4 = \mathbf{E}_x\left[x \otimes x \otimes x \otimes x\right] \tag{40}$$

$$= \mathbf{E}_z[G\left(z \odot y\right) \otimes G\left(z \odot y\right) \otimes G\left(z \odot y\right) \otimes G\left(z \odot y\right)]$$
$$+ \mathbf{E}_z[\mathfrak{S}_6\left[G\left(z \odot y\right) \otimes G\left(z \odot y\right) \otimes \epsilon \otimes \epsilon\right]] + \mathbf{E}\left[\epsilon \otimes \epsilon \otimes \epsilon \otimes \epsilon\right]$$

$$= T\left(\mathbf{E}_z\left[z \otimes z \otimes z \otimes z\right] \odot \mathbf{E}_y\left[y^4\right]\mathbf{1}, A, A, A, A\right) + \sigma^2\mathfrak{S}_6\left[S_2 \otimes \mathbf{1}\right] + \sigma^4\mathfrak{S}_3\left[\mathbf{1} \otimes \mathbf{1}\right] \tag{41}$$

$$= \mathfrak{S}_3\left[S_2 \otimes S_2\right] + T\left(\mathrm{diag}\left(\mathbf{E}_y\left[y^4\right]\pi_i - 3\mathbf{E}_y\left[y^2\right]^2 \pi_i^2\right), A, A, A, A\right)$$
$$+ \sigma^2\mathfrak{S}_6\left[S_2 \otimes \mathbf{1}\right] + \sigma^4\mathfrak{S}_3\left[\mathbf{1} \otimes \mathbf{1}\right] \tag{42}$$

$$= \mathfrak{S}_3\left[S_2 \otimes S_2\right] + T\left(3\left(\pi_i - \pi_i^2\right), A, A, A, A\right) + \sigma^2\mathfrak{S}_6\left[S_2 \otimes \mathbf{1}\right] + m_4\mathfrak{S}_3\left[\mathbf{1} \otimes \mathbf{1}\right] \tag{43}$$

where $m_4$ can be inferred by (17).

If the prior on $Y$ is drawn from a Laplace distribution the model is called a iICA. The lower-order moments are similar to that of isFA, except for $\mathbf{E}_y\left[y^2\right] = 2$ and $\mathbf{E}_y\left[y^4\right] = 24$. Replacing these terms in Equation (39) and (42) yields the claim.

# B    Concentration measure of bounded moments

## B.1    Proof of the Moment Bound of Theorem 2

Denote by $X$ the $m$-sample used in generating $\hat{M}_l$. Moreover, denote by

$$\Xi[X] := \sup_{u:\|u\| \leq 1}\left|T[M_l, u, \cdots, u] - T[\hat{M}_l, u, \cdots, u]\right| \tag{44}$$

the largest deviation between empirical and expected moments, when applied to the test vectors $u$. Bounding this quantity directly is desirable since it allows us to avoid having to derive *pointwise* bounds with regard to $M_l$. We prove that $\Xi[X]$ is concentrated using McDiarmid's bound [25]. Substituting single observations in $\Xi[X]$ yields

$$\left|\Xi[X] - \Xi[(X \setminus \{x_j\}) \cup \{x'\}]\right| \leq \frac{1}{m}\left[T\left[\phi_l(x_j) - \phi_l(x'), u, \ldots u\right]\right] \tag{45}$$

$$\leq \frac{1}{m}\left[\|\phi(x_j)\|^l + \|\phi(x')\|^l\right] \leq \frac{2}{m}R^l. \tag{46}$$

Plugging the bound of $2R^l/m$ into McDiarmid's theorem shows that the random variable $\Xi[X]$ is concentrated for $\Pr\{\Xi[X] - \mathbf{E}_X[\Xi[X]] > \epsilon\} \leq \delta$ with probability $\delta \leq \exp\left(-\frac{m\epsilon^2}{2R^{2l}}\right)$. Solving the bound for $\epsilon$ shows that with probability at least $1 - \delta$ we have that $\epsilon \leq \sqrt{-2\log\delta/m}R^i$.

The next step is to bound the expectation of $\Xi[X]$. For this we exploit the ghost sample trick and the convexity of expectations. This leads to the following:

$$\mathbf{E}_X\left[\Xi[X]\right] \leq \mathbf{E}_{X,X'}\left[\sup_{u:\|u\| \leq 1}\left|T[M_l, u, \cdots, u] - T[\hat{M}_l, u, \cdots, u]\right|\right]$$

$$= \mathbf{E}_\sigma \mathbf{E}_{X,X'}\left[\sup_{u:\|u\| \leq 1}\left|\frac{1}{m}\sum_{j=1}^m \sigma_j\left(T[\phi_l(x_j), u, \cdots, u] - T[\phi_l(x'_j)u, \cdots, u]\right)\right|\right]$$

$$\leq \frac{2}{m}\mathbf{E}_\sigma \mathbf{E}_X\left[\sup_{u:\|u\| \leq 1}\left|\sum_{j=1}^m \sigma_j T[\phi_l(x_j), u, \cdots, u]\right|\right] \tag{47}$$

$$\leq \frac{2}{m}\mathbf{E}_\sigma \mathbf{E}_X\left[\left\|\sum_{j=1}^m \sigma_j \phi_l(x_j)\right\|\right] \leq \frac{2}{m}\mathbf{E}_X\left[\mathbf{E}_\sigma\left[\left\|\sum_{j=1}^m \sigma_j \phi_l(x_j)\right\|^2\right]\right]^{\frac{1}{2}} \leq \frac{2R^l}{\sqrt{m}} \tag{48}$$

Here the first inequality follows from convexity of the argument. The subsequent equality is a consequence of the fact that $X$ and $X'$ are drawn from the same distribution, hence a swapping permutation with the ghost-sample leaves terms unchanged; The following inequality is an application of the triangle inequality. Next we use the Cauchy-Schwartz inequality, convexity and last the fact that $\|\phi(x)\| \leq R$. Combining both bounds yields $\epsilon \leq \left[2 + \sqrt{-2\log\delta}\right] R^l / \sqrt{m}$.

## B.2 Tools for bounding tensors with bounded moments

To prove the guarantees for tensors, we rely on the triangle inequality on tensorial reductions

$$\sup_u \left| T(A+B, u) - T(A'+B', u) \right| \leq \sup_u \left| T(A, u) - T(A', u) \right| + \sup_u \left| T(B, u) - T(B', u) \right| \quad (49)$$

and moreover, the fact that for products of bounded random variables the guarantees are additive, as stated in the lemma below:

**Lemma 4** *Denote by $f_i$ random variables and by $\hat{f}_i$ their estimates. Moreover, assume that each of them is bounded via $|f_i| \leq R_i$ and $|\hat{f}_i| \leq R_i$ and*

$$\Pr\left\{ |\mathbf{E}[f_i] - \hat{f}_i| > \epsilon_i \right\} \leq \delta_i. \quad (50)$$

*In this case the product is bounded via*

$$\Pr\left\{ \left| \prod_i \mathbf{E}[f_i] - \prod_i \hat{f}_i \right| > \epsilon \right\} \leq \sum_i \delta_i \quad \text{where } \epsilon = \left[ \prod_i R_i \right] \left[ \sum_i \frac{\epsilon_i}{R_i} \right] \quad (51)$$

**Proof** We prove the claim for two variables, say $f_1$ and $f_2$. We have

$$\left| \mathbf{E}[f_1]\mathbf{E}[f_2] - \hat{f}_1\hat{f}_2 \right| \leq \left| (\mathbf{E}[f_1] - \hat{f}_1)\mathbf{E}[f_2] \right| + \left| \hat{f}_1(\mathbf{E}[f_2] - \hat{f}_2) \right| \leq \epsilon_1 R_2 + R_1 \epsilon_2$$

with probability at least $1 - \delta_1 - \delta_2$, when applying the union bound over $\mathbf{E}[f_1] - \hat{f}_1$ and $\mathbf{E}[f_2] - \hat{f}_2$ respectively. Rewriting terms yields the claim for $n = 2$. To see the claim for $n > 2$ simply use the fact that we can decompose the bound into a chain of inequalities involving exactly one difference, say $\mathbf{E}[f_i] - \hat{f}_i$ and $n - 1$ instances of $\mathbf{E}[f_j]$ or $\hat{f}_j$ respectively. We omit details since they are straightforward to prove (and tedious). ∎

## C  Proof of Equation (21)

For simplicity in the proof, in Equation (13) (14) (15), we define the diagonal coefficients for $S_i$ to be $C_i \in \mathbb{R}^K$, i.e., $C_2 = \pi - \pi^2$, $C_3 = \pi - 3\pi^2 + 2\pi^3$ and $C_4 = \pi - 7\pi^2 + 12\pi^3 - 6\pi^4$, so that

$$S_2 = T(\operatorname{diag}(C_2), A, A), \quad S_3 = T(\operatorname{diag}(C_3), A, A, A), \quad S_4 = T(\operatorname{diag}(C_4), A, A, A, A).$$

Following step 6 in Algorithm 1, we obtain whitening matrix $W$ by doing svd on $S_2$. Suppose the svd of matrix $T(\operatorname{diag}(\sqrt{C_2}), A) = U\Sigma^{1/2}V^\top$, we have $S_2 = U\Sigma^{1/2}V^\top V\Sigma^{1/2}U^\top = USU^T$ and $W = U\Sigma^{-1/2}$. Using the fact that $S_3 = T\left(\operatorname{diag}\left(C_3 C_2^{-3/2}\right), \operatorname{diag}(\sqrt{C_2})A, \operatorname{diag}(\sqrt{C_2})A, \operatorname{diag}(\sqrt{C_2})A\right)$, we have

$$
\begin{aligned}
W_3 &= T(S_3, W, W, W) \\
&= T\left(\operatorname{diag}\left(C_3 C_2^{-3/2}\right), \Sigma^{-1/2}U^\top(U\Sigma^{1/2}V^\top), \Sigma^{-1/2}U^\top(U\Sigma^{1/2}V^\top), \Sigma^{-1/2}U^\top(U\Sigma^{1/2}V^\top)\right) \\
&= T\left(\operatorname{diag}\left(C_3 C_2^{-3/2}\right), V^\top, V^\top, V^\top\right).
\end{aligned}
\quad (52)
$$

The diagonalized tensor $W_3$, with some permutation $\tau$ on $[K]$ and $s_i \in \{\pm 1\}$, has eigenvalues and eigenvectors:

$$\lambda_i = s_i C_{3,i} C_{2,i}^{-3/2}, \quad v_i = s_i(V^\top), e_{\tau(i)}, \quad (53)$$

where $C_{i,j}$ representing the $j$-th element in $C_i$. After obtaining $v_i$, we multiply $v_i$ by $(W^\dagger)^\top$ to rotate it back to $A_i$ as describing in step 15 in Algorithm 1, where $W^\dagger = (W^\top W)^{-1}W^\top = \Sigma^{1/2}U^\top$, we get

$$(W^\dagger)^\top v_i = s_i U\Sigma^{1/2}V^\top e_{\tau(i)} = s_i T(\operatorname{diag}(\sqrt{C_2}), A)e_{\tau(i)} = s_i\sqrt{C_{2,i}}A_{\tau(i)}, \quad (54)$$

which yields $A_{\tau(i)} = \frac{(W^\dagger)^\top v_i}{s_i\sqrt{C_{2,i}}}$. With the fact that $s_i = C_{3,i}C_{2,i}^{-3/2}\lambda_i^{-1}$ from Equation (53), we have

$$A_{\tau(i)} = \frac{\lambda_i}{\left(C_{3,i}C_{2,i}^{-1}\right)}(W^\dagger)^\top v_i = C_{2,i}^{-1/2}(W^\dagger)^\top v_i. \tag{55}$$

Plug in the definition of $C_2$, we get the scale factor for $i \in [K_1]$. For $A_i$ which are recovered by conducting tensor decomposition on $W_4$, we first examine

$$W_4 = T(S_4, W, W, W, W) = T\left(\text{diag}\left(C_{4,i}C_{2,i}^{-2}\right), V^\top, V^\top, V^\top, V^\top\right), \tag{56}$$

and obtain

$$\lambda_i = C_{4,i}C_{2,i}^{-2}, \quad v_i = s_i(V^\top)e_{\tau(i)}. \tag{57}$$

By using the fact that $s_i = s_iC_{4,i}C_{2,i}^{-2}\lambda_i^{-1}$ and Equation (54), we have

$$A_{\tau(i)} = \frac{(W^\dagger)^\top v_i}{s_i\sqrt{C_{2,i}}} = \frac{s_i}{\left(C_{4,i}C_{2,i}^{-3/2}\right)\lambda_i^{-1}} = s_i(W^\dagger)^\top v_i = s_iC_{2,i}^{-1/2}(W^\dagger)^\top v_i, \quad \forall i \in [K_1 + 1, \cdots, K]. \tag{58}$$

Note that the value of $\pi_i$ used to construct $C_j$ can be recovered by Equation (53) and (57) after obtaining $\lambda_i$.

# D   Reconstruction accuracy for Algorithm 1

In this section, we provides bounds for moments of linear gaussian latent feature model. The concentration behavior is more complicated than that of the bounded moments in Theorem 2 due to the additive Gaussian noise. Here we restate the model as

$$x = Az + \epsilon \tag{59}$$

where $x \in \mathbb{R}^d$ is the observation, $z \in \{0,1\}^K$ is a binary vector indicating the possession of certain latent vector and $\epsilon$ is gaussian noise drawn from $N(0, \sigma^2 \mathbf{1})$.

## D.1   Concentration measure of unbounded moments

In order to utilize the bounds for gaussian random vectors, we need to subtract the term $Az$ from $x$ by operating $[\hat{M} - M]$. The bounds for observation generated by different $z$ are examined separately. Let $B = \{x_1, x_2, \cdots, x_n\}$ and, for a specific $z_i \in \{0,1\}^K$, write $B_{z_i} := \{x \in B : z = z_i\}$ and $\hat{w}_{z_i} = |B_{z_i}|/|B|$ for $i \in \{0, 1 \cdots 2^K - 1\}$ and $z_i = \text{binary}(i)$. Define the conditional moments to be

$$M_{1,z_i} := \mathbf{E}[x|z = z_i], \quad M_{2,z_i} := \mathbf{E}[x \otimes x|z = z_i], \quad M_{3,z_i} := \mathbf{E}[x \otimes x \otimes x|z = z_i],$$
$$M_{4,z_i} := \mathbf{E}[x \otimes x \otimes x \otimes x|z = z_i],$$

while the empirical moments are

$$\hat{M}_{1,z_i} := |B_{z_i}|^{-1}\sum_{x \in B_{z_i}} x, \quad \hat{M}_{2,z_i} := |B_{z_i}|^{-1}\sum_{x \in B_{z_i}} x \otimes x, \quad \hat{M}_{3,z_i} := |B_{z_i}|^{-1}\sum_{x \in B_{z_i}} x \otimes x \otimes x,$$
$$\hat{M}_{4,z_i} := |B_{z_i}|^{-1}\sum_{x \in B_{z_i}} x \otimes x \otimes x \otimes x.$$

**Lemma 5** *(Concentration of conditional empirical moments) With probability greater than $1 - \delta$, pick any $\delta \in (0, 1)$ and any random matrix $V \in \mathbb{R}^{d \times r}$ of rank $r$, the following guarantee holds*
*1. For the first-order moments, we have*

$$\left\|T\left(\hat{M}_{1,z_i} - M_{1,z_i}, V\right)\right\|_2 \le \sigma\|V\|_2\sqrt{\frac{r + 2\sqrt{r\ln\left(2^K/\delta\right)} + 2\ln\left(2^K/\delta\right)}{\hat{w}_{z_i}n}}, \quad \forall i \in \left\{0, 1 \cdots 2^K - 1\right\}$$

*2. For the second-order moments, we have*

$$\left\|T\left(\hat{M}_{2,z_i} - M_{2,z_i}, V, V\right)\right\|_2$$
$$\le \sigma^2\|V\|_2^2\left(\sqrt{\frac{128\left(r\ln 9 + \ln\left(2^{K+2}/\delta\right)\right)}{\hat{w}_{z_i}n}} + \frac{4\left(r\ln 9 + \ln\left(2^{K+2}/\delta\right)\right)}{\hat{w}_{z_i}n}\right)$$
$$+ 2\sigma\left\|V^\top M_{1,z_i}\right\|_2\|V\|_2\sqrt{\frac{r + 2\sqrt{r\ln\left(2^{K+1}/\delta\right)} + 2\ln\left(2^{K+1}/\delta\right)}{\hat{w}_{z_i}n}}, \quad \forall i \in \left\{0, 1 \cdots 2^K - 1\right\}$$

*3. For the third-order moments, we have*

$$\left\| T\left(\hat{M}_{3,z_i} - M_{3,z_i}, V, V, V\right)\right\|_2 \le \sigma^3 \left\| V\right\|_2^3 \left(\sqrt{\frac{108 e^3 \lceil r \ln 13 + \ln\left(3 \cdot 2^K/\delta\right)\rceil^3}{\hat{w}_{z_i} n}}\right)$$

$$+ 3\sigma^2 \left\| V^\top M_{1,z_i}\right\|_2 \left\| V\right\|_2^2 \left(\sqrt{\frac{128\left(r \ln 9 + \ln\left(3 \cdot 2^{K+1}/\delta\right)\right)}{\hat{w}_{z_i} n}} + \frac{4\left(r \ln 9 + \ln\left(3 \cdot 2^{K+1}/\delta\right)\right)}{\hat{w}_{z_i} n}\right)$$

$$+ 3\sigma \left\| V^\top M_{1,z_i}\right\|_2^2 \left\| V\right\|_2 \sqrt{\frac{r + 2\sqrt{r \ln\left(3 \cdot 2^K/\delta\right)} + 2\ln\left(3 \cdot 2^K/\delta\right)}{\hat{w}_{z_i} n}}, \;\; \forall i \in \left\{0, 1 \cdots 2^K - 1\right\}$$

*4. For the fourth-order moments, we have*

$$\left\| T\left(\hat{M}_{4,z_i} - M_{4,z_i}, V, V, V, V\right)\right\|_2$$

$$\le \sigma^4 \left\| V\right\|_2^4 \left(\sqrt{\frac{8192\left(r \ln 17 + \ln\left(4 \cdot 2^{K+1}/\delta\right)\right)^2}{n^2}} + \frac{32\left(r \ln 17 + \ln\left(4 \cdot 2^{K+1}/\delta\right)\right)^3}{n^3}\right)$$

$$+ 4\sigma^3 \left\| V^\top M_{1,z_i}\right\|_2 \left\| V\right\|_2^3 \left(\sqrt{\frac{108 e^3 \lceil r \ln 13 + \ln\left(4 \cdot 2^K/\delta\right)\rceil^3}{\hat{w}_{z_i} n}}\right)$$

$$+ 6\sigma^2 \left\| V^\top M_{1,z_i}\right\|_2^2 \left\| V\right\|_2^2 \left(\sqrt{\frac{128\left(r \ln 9 + \ln\left(4 \cdot 2^{K+1}/\delta\right)\right)}{\hat{w}_{z_i} n}} + \frac{4\left(r \ln 9 + \ln\left(4 \cdot 2^{K+1}/\delta\right)\right)}{\hat{w}_{z_i} n}\right)$$

$$+ 4\sigma \left\| V^\top M_{1,z_i}\right\|_2^3 \left\| V\right\|_2 \sqrt{\frac{r + 2\sqrt{r \ln\left(4 \cdot 2^K/\delta\right)} + 2\ln\left(4 \cdot 2^K/\delta\right)}{\hat{w}_{z_i} n}}, \;\; \forall i \in \left\{0, 1 \cdots 2^K - 1\right\},$$

**Proof** Here we only show the derivation of the fourth-order conditional moments. The other inequalities can be found in [23]. Under the stated model, the fourth-order conditional moment can be expended as

$$M_{4,z_i} = M_{1,z_i} \otimes M_{1,z_i} \otimes M_{1,z_i} \otimes M_{1,z_i} + \sigma^2 \mathfrak{S}_6 \left[M_{1,z_i} \otimes M_{1,z_i} \otimes \mathbf{1}\right] + \mathbf{E}\left[\epsilon \otimes \epsilon \otimes \epsilon \otimes \epsilon\right],$$

which yields

$$\hat{M}_{4,z_i} - M_{4,z_i}$$

$$= \frac{1}{\hat{w}_{z_i} n}\left(\sum_{x \in B_{z_i}} (x_j - M_{1,z_i}) \otimes (x_j - M_{1,z_i}) \otimes (x_j - M_{1,z_i}) \otimes (x_j - M_{1,z_i}) - \sigma^4 \mathfrak{S}_3 \left[\mathbf{1} \otimes \mathbf{1}\right]\right.$$

$$+ \sum_{x \in B_{z_i}} \left(\mathfrak{S}_4 \left[M_{1,z_i} \otimes (x_j - M_{1,z_i}) \otimes (x_j - M_{1,z_i}) \otimes (x_j - M_{1,z_i})\right]\right)$$

$$+ \sum_{x \in B_{z_i}} \left(\mathfrak{S}_6 \left[M_{1,z_i} \otimes M_{1,z_i} \otimes \left((x_j - M_{1,z_i}) \otimes (x_j - M_{1,z_i}) - \sigma^2 \mathbf{1}\right)\right]\right)$$

$$+ \sum_{x \in B_{z_i}} \mathfrak{S}_4 \left[M_{4,z_i} \otimes M_{4,z_i} \otimes M_{4,z_i} \otimes (x_j - M_{1,z_i})\right]\right) \tag{60}$$

Suppose $V = V_1 \Sigma V_2^\top$ is the SVD of $V$, where $V_1 \in \mathrm{R}^{d \times r}$ consists of orthonormal columns. With $y_{j,z_i} = V_1^\top (x_j - M_{1,z_i})$, applying triangle inequalities to Equation (60) yields

$$\left\| T\left(\hat{M}_{4,z_i} - M_{4,z_i}, V, V, V, V\right)\right\|_2$$

$$\le \left\| V\right\|_2^4 \left\|\frac{1}{\hat{w}_{z_i} n}\sum_{x \in B_{z_i}} \left(y_{j,z_i} \otimes y_{j,z_i} \otimes y_{j,z_i} \otimes y_{j,z_i} - \sigma^4 \mathfrak{S}_3 \left[\mathbf{1} \otimes \mathbf{1}\right]\right)\right\|_2$$

$$+ 4\left\| V^\top M_{1,z_i}\right\|_2 \left\|\frac{1}{\hat{w}_{z_i} n}\sum_{x \in B_{z_i}} y_{j,z_i} \otimes y_{j,z_i} \otimes y_{j,z_i}\right\|_2$$

$$+ 6\left\| V^\top M_{1,z_i}\right\|_2^2 \left\|\frac{1}{\hat{w}_{z_i} n}\sum_{x \in B_{z_i}} \left(y_{j,z_i} \otimes y_{j,z_i} - \sigma^2 \mathbf{1}\right)\right\|_2 + 4\left\| R^\top M_{1,z_i}\right\|_2^3 \left\|\frac{1}{\hat{w}_{z_i} n}\sum_{x \in B_{z_i}} y_{j,z_i}\right\|_2,$$

By using Lemma 13, we bound the first term by

$$\Pr\left[\left\|\frac{1}{\hat{w}_{z_i}n}\sum_{x\in B_{z_i}}\left(y_{j,z_i}\otimes y_{j,z_i}\otimes y_{j,z_i}\otimes y_{j,z_i}-\mathbf{E}\left[\epsilon\otimes\epsilon\otimes\epsilon\otimes\epsilon\right]\right)\right\|_2\right.$$

$$\left.>\sigma^4\sqrt{\frac{8192\left(r\ln 17+\ln\left(2K/\delta\right)\right)^2}{n^2}+\frac{32\left(r\ln 17+\ln\left(2K/\delta\right)\right)^3}{n^3}}\right]\leq\delta. \tag{61}$$

∎

The other norm can be bounded by using the bounds for low-order conditional moments. We finish the proof by adding the bounds for every term. By using inequalities for conditional moments, We get the bounds for completed moments stated in the following Lemma.

**Lemma 6** *( Lemma 6 in [23]; Concentration of empirical moments) For a fixed matrix $V\in\mathbb{R}^{d\times r}$,*

$$\left\|T\left(\hat{M}_i-M_i,V,\cdots,V\right)\right\|_2$$
$$\leq\left(1+2^{K/2}\epsilon_w\right)\max_{z_j}\left\|T\left(\hat{M}_{i,z_j}-M_{i,z_j},V,\cdots,V\right)\right\|_2+2^{K/2}\max_{z_j}\left\|T\left(M_{i,z_j},V,\cdots,V\right)\right\|_2\varepsilon_w$$
$$\forall i\in[4],\forall j\in\left\{0,1\cdots 2^K-1\right\}$$

*where $\varepsilon_w=\left(\sum_{z_j}\left(\hat{w}_{z_j}-w_{z_j}\right)^2\right)^{\frac{1}{2}}\leq\frac{1+\sqrt{\ln(1/\delta)}}{\sqrt{n}}$.*

## D.2 Estimation of $\sigma$, $S_2$, $S_3$, $S_4$

Note that we have $\sigma^2=\lambda_{min}\left[M_2-M_1\otimes M_1\right]=\varsigma_K[M_2-M_1\otimes M_1]$, where $\varsigma_t\left[M\right]$ denoting the $t-$ th singular value of matrix $M$ which is defined in Theorem 3. Here we define $\hat{S}_{2,K}$ to be the best rank $k$ approximation of $\hat{M}_2-\hat{M}_1\otimes\hat{M}_1-\hat{\sigma}^2\mathbf{1}$, which is the truncated matrix $S_2$ in Algorithm 1. $\hat{S}_i$ denotes the empirical tensors derived from summation of $\hat{M}_i$ and $\hat{\sigma}$. $S_i$ denotes the theoretical values.

**Lemma 7** *(Accuracy of $\sigma^2$, $\sigma^4$ and $M_{2,K}$)*

$$\left|\hat{\sigma}^2-\sigma^2\right|\leq\left\|\hat{M}_2-M_2\right\|_2+\left\|\hat{M}_1-M_1\right\|_2^2+2\left\|\hat{M}_1-M_1\right\|_2\|M_1\|_2 \tag{62}$$

$$\left|\hat{\sigma}^4-\sigma^4\right|\leq\left|\hat{\sigma}^2-\sigma^2\right|^2+2\sigma^2\left|\hat{\sigma}^2-\sigma^2\right| \tag{63}$$

$$\left\|\hat{S}_{2,k}-S_2\right\|_2\leq 4\left(\left\|\hat{M}_2-M_2\right\|_2+\left\|\hat{M}_1-M_1\right\|_2^2+2\|M_1\|_2\left\|\hat{M}_1-M_1\right\|_2\right) \tag{64}$$

**Proof**

For the first order tensor, the inequality holds trivially due to the guarantees for $\left\|\hat{M}_1-M_1\right\|_2$. Next we bound the difference in variance estimates. Using the fact that differences in the $k$-th eigenvalues are bounded by the matrix norm of the difference we have that

$$\left|\hat{\sigma}^2-\sigma^2\right|=\left|\varsigma_k\left[\hat{M}_2-\hat{M}_1\otimes\hat{M}_1\right]-\varsigma_k[M_2-M_1\otimes M_1]\right| \tag{65}$$

$$\leq\left\|\left[\hat{M}_2-\hat{M}_1\otimes\hat{M}_1\right]-[M_2-M_1\otimes M_1]\right\|_2 \tag{66}$$

$$\leq\left\|\hat{M}_2-M_2\right\|_2+\left\|\hat{M}_1-M_1\right\|_2^2+2\left\|\hat{M}_1-M_1\right\|_2\|M_1\|_2. \tag{67}$$

The second inequality follows the Weyl's inequality and the last inequality is obtained by the triangle inequality. For estimation of $\hat{\sigma}^4$,

$$\left|\hat{\sigma}^4-\sigma^4\right|\leq\left|(\hat{\sigma}^2-\sigma^2)^2+2\sigma^2(\hat{\sigma}^2-\sigma^2)\right|\leq\left|\hat{\sigma}^2-\sigma^2\right|^2+2\sigma^2\left|\hat{\sigma}^2-\sigma^2\right|. \tag{68}$$

For the last claimed inequality, with Weyl's inequality,

$$\left\|\hat{S}_{2,k}-\left(\hat{M}_2-\hat{M}_1\otimes\hat{M}_1-\hat{\sigma}^2\mathbf{1}\right)\right\|_2\leq\varsigma_{k+1}\left[\hat{M}_2-\hat{M}_1\otimes\hat{M}_1-\hat{\sigma}^2\mathbf{1}\right] \tag{69}$$

$$=\left\|\varsigma_{k+1}\left[\hat{M}_2-\hat{M}_1\otimes\hat{M}_1-\hat{\sigma}^2\mathbf{1}\right]-\varsigma_{k+1}\left[M_2-M_1\otimes M_1-\sigma^2\mathbf{1}\right]\right\|_2 \tag{70}$$

$$\leq\left\|\hat{M}_2-\hat{M}_1\otimes\hat{M}_1-\hat{\sigma}^2\mathbf{1}-\left(M_2-M_1\otimes M_1-\sigma^2\mathbf{1}\right)\right\|_2 \tag{71}$$

, which yields

$$\left\| \hat{S}_{2,k} - S_2 \right\|_2 \leq \left\| \hat{S}_{2,k} - \left( \hat{M}_2 - \hat{M}_1 \otimes \hat{M}_1 - \hat{\sigma}^2 \mathbf{1} \right) \right\|_2$$
$$+ \left\| \hat{M}_2 - \hat{M}_1 \otimes \hat{M}_1 - \hat{\sigma}^2 \mathbf{1} - \left( M_2 - M_1 \otimes M_1 - \sigma^2 \mathbf{1} \right) \right\|_2 \tag{72}$$

$$\leq 2 \left( \left\| \hat{M}_2 - M_2 \right\|_2 + \left\| \hat{M}_1 - M_1 \right\|_2^2 + 2 \left\| M_1 \right\|_2 \left\| \hat{M}_1 - M_1 \right\|_2 + |\hat{\sigma}^2 - \sigma^2| \right) \tag{73}$$

$$\leq 4 \left( \left\| \hat{M}_2 - M_2 \right\|_2 + \left\| \hat{M}_1 - M_1 \right\|_2^2 + 2 \left\| M_1 \right\|_2 \left\| \hat{M}_1 - M_1 \right\|_2 \right). \tag{74}$$

∎

The inequalities for $\sigma$ can be used for bounding the tensors $S_2$, $S_3$ and $S_4$, which will be shown next, and the inequality for $S_{2,k}$ will be used in bounding whitened tensor in Section D.3.

**Lemma 8** *(Accuracy of $S_2$, $S_3$ and $S_4$) For a fixed matrix $V \in \mathbb{R}^{d \times K}$*

$$\left\| T \left( \hat{S}_2 - S_2, V, V \right) \right\|_2 \leq \left\| T \left( \hat{M}_2 - M_2, V, V \right) \right\|_2 + \left\| T \left( \hat{M}_1 - M_1, V \right) \right\|_2^2$$
$$+ 2 \left\| T \left( M_1, V \right) \right\|_2 \left\| T \left( \hat{M}_1 - M_1, V \right) \right\|_2 + \|V\|_2^2 |\hat{\sigma}^2 - \sigma^2| \tag{75}$$

$$\left\| T \left( \hat{S}_3 - S_3, V, V, V \right) \right\|_2$$
$$\leq \left\| T \left( \hat{M}_3 - M_3, V, V, V \right) \right\|_2 + \left( \left\| T \left( \hat{M}_1 - M_1, V \right) \right\|_2 + \| T \left( M_1, V \right) \|_2 \right)^3 - \| T \left( M_1, V \right) \|_2^3$$
$$+ 3 \left( \left\| T \left( \hat{M}_1 - M_1, V \right) \right\|_2 \left\| T \left( \hat{S}_2 - S_2, V, V \right) \right\|_2 + \| T \left( M_1, V \right) \|_2 \left\| T \left( \hat{S}_2 - S_2, V, V \right) \right\|_2 \right.$$
$$+ \left. \left\| T \left( \hat{M}_1 - M_1, V \right) \right\|_2 \| T \left( S_2, V, V \right) \|_2 \right) + 3 \|V\|_2^2 \left( |\hat{\sigma}^2 - \sigma^2| \left\| T \left( \hat{M}_1 - M_1, V \right) \right\|_2 \right.$$
$$+ \sigma^2 \left\| T \left( \hat{M}_1 - M_1, V \right) \right\|_2 + |\hat{\sigma}^2 - \sigma^2| \| T \left( M_1, V \right) \|_2 \bigg) \tag{76}$$

$$\left\| T \left( \hat{S}_4 - S_4, V, VV, V \right) \right\|_2$$
$$\leq \left\| T \left( \hat{M}_4 - M_4, V, V, V, V \right) \right\|_2 + \left( \left\| T \left( \hat{M}_1 - M_1, V \right) \right\|_2 + \| T \left( M_1, V \right) \|_2 \right)^4 - \| T \left( M_1, V \right) \|_2^4$$
$$+ 6 \left\| T \left( \hat{S}_2 - S_2, V, V \right) \right\|_2 \| T \left( M_1, V \right) \|_2^2 + 6 \left( \left\| T \left( \hat{S}_2 - S_2, V, V \right) \right\|_2 + \| T \left( S_2, V, V \right) \|_2 \right)$$
$$\left( 2 \| T \left( M_1, V \right) \|_2 \left\| T \left( \hat{M}_1 - M_1, V \right) \right\|_2 + \left\| T \left( \hat{M}_1 - M_1, V \right) \right\|_2^2 \right) + 3 \left( \left\| T \left( \hat{S}_2 - S_2, V, V \right) \right\|_2^2 \right.$$
$$+ 2 \left\| T \left( \hat{S}_2 - S_2, V, V \right) \right\|_2 \| T \left( S_2, V, V \right) \|_2 \right) + 6 \|V\|_2^2 \left( \sigma^2 \left\| T \left( \hat{S}_2 - S_2, V, V \right) \right\|_2 + \right.$$
$$+ |\hat{\sigma}^2 - \sigma^2| \left( \left\| T \left( \hat{S}_2 - S_2, V, V \right) \right\|_2 + \| T \left( S_2, V, V \right) \|_2 \right) \right) + 3 |\hat{\sigma}^4 - \sigma^4| \|V\|_2^4$$
$$+ 4 \left( \left\| T \left( \hat{S}_3 - S_3, V, V, V \right) \right\|_2 \left\| T \left( \hat{M}_1 - M_1, V \right) \right\|_2 + \| T \left( M_1, V \right) \|_2 \left\| T \left( \hat{S}_3 - S_3, V, V, V \right) \right\|_2 \right.$$
$$+ \| T \left( S_3, V, V, V \right) \|_2 \left\| T \left( \hat{M}_1 - M_1, V \right) \right\|_2 \bigg) \tag{77}$$

**Proof** To bound the second order tensor, we use the inequality for bounding $\hat{\sigma}$ in Lemma 7 and get

$$\left\| T \left( \hat{S}_2, V, V \right) - T \left( S_2, V, V \right) \right\|_2$$
$$\leq \left\| T \left( \hat{M}_2 - M_2, V, V \right) \right\|_2 + \left\| T \left( (\hat{M}_1 - M_1) \otimes (\hat{M}_1 - M_1), V, V \right) \right\|_2 \tag{78}$$
$$+ 2 \left\| T \left( M_1 \otimes (\hat{M}_1 - M_1), V, V \right) \right\|_2 + \|V\|_2^2 |\hat{\sigma}^2 - \sigma^2|$$
$$\leq \left\| T \left( \hat{M}_2 - M_2, V, V \right) \right\|_2 + \left\| T \left( \hat{M}_1 - M_1, V, V \right) \right\|_2^2 + 2 \| T \left( M_1, V \right) \|_2 \left\| T \left( \hat{M}_1 - M_1, V, V \right) \right\|_2$$
$$+ \|V\|_2^2 |\hat{\sigma}^2 - \sigma^2|. \tag{79}$$

Similarly, for $\hat{S}_3$, we have that

$$
\left\| T\left(\hat{S}_3, V, V, V\right) - T\left(S_3, V, V, V\right) \right\|_2
$$
$$
\leq \left\| T\left(\hat{M}_3 - M_3, V, V, V\right) \right\|_2 + \left\| T\left(\hat{M}_1 \otimes \hat{M}_1 \otimes \hat{M}_1 - M_1 \otimes M_1 \otimes M_1, V, V, V\right) \right\|_2
$$
$$
+ 3\left\| T\left(\hat{S}_1 \otimes \hat{S}_2 - S_1 \otimes S_2, V, V, V\right) \right\|_2 + 3\left\| T\left(\left(\hat{\sigma}^2 \hat{M}_1 - \sigma^2 M_1\right) \otimes \mathbf{1}, V, V, V\right) \right\|_2. \tag{80}
$$

Note that the second term can be written as

$$
\hat{M}_1 \otimes \hat{M}_1 \otimes \hat{M}_1 - M_1 \otimes M_1 \otimes M_1
$$
$$
= \left(\hat{M}_1 - M_1\right) \otimes \left(\hat{M}_1 - M_1\right) \otimes \left(\hat{M}_1 - M_1\right)
$$
$$
+ \mathfrak{S}_3\left[M_1 \otimes \left(\hat{M}_1 - M_1\right) \otimes \left(\hat{M}_1 - M_1\right)\right] + \mathfrak{S}_3\left[M_1 \otimes M_1 \otimes \left(\hat{M}_1 - M_1\right)\right]. \tag{81}
$$

Using the same expansion trick, the third term becomes

$$
\hat{S}_1 \otimes \hat{S}_2 - S_1 \otimes S_2 = (\hat{S}_1 - S_1) \otimes (\hat{S}_2 - S_2) + S_1 \otimes (\hat{S}_2 - S_2) + (\hat{S}_1 - S_1) \otimes S_2. \tag{82}
$$

Using triangle inequality, the bound for Equation (81) is

$$
\left\| T\left(\hat{M}_1 \otimes \hat{M}_1 \otimes \hat{M}_1 - M_1 \otimes M_1 \otimes M_1, V, V, V\right) \right\|_2
$$
$$
\leq \left\| T\left(\hat{M}_1 - M_1, V\right) \right\|_2^3 + 3\left\| T\left(M_1, V\right) \right\|_2 \left\| T\left(\hat{M}_1 - M_1, V\right) \right\|_2^2
$$
$$
+ 3\left\| T\left(M_1, V\right) \right\|_2^2 \left\| T\left(\hat{M}_1 - M_1, V\right) \right\|_2, \tag{83}
$$

and the bound for Equation (82) is

$$
\left\| T\left(\hat{S}_1 \otimes \hat{S}_2 - S_1 \otimes S_2, V, V, V\right) \right\|_2
$$
$$
\leq \left\| T\left(\hat{S}_1 - S_1, V\right) \right\|_2 \left\| T\left(\hat{S}_2 - S_2, V, V\right) \right\|_2 + \left\| T\left(S_1, V\right) \right\|_2 \left\| T\left(\hat{S}_2 - S_2, V, V\right) \right\|_2
$$
$$
+ \left\| T\left(\hat{S}_1 - S_1, V\right) \right\|_2 \left\| T\left(S_2, V, V\right) \right\|_2 \tag{84}
$$
$$
\left\| T\left(\left(\hat{\sigma}^2 \hat{M}_1 - \sigma^2 M_1\right) \otimes \mathbf{1}, V, V\right) \right\|_2
$$
$$
\leq \|V\|_2^2 \left(|\hat{\sigma}^2 - \sigma^2| \left\| T\left(\hat{M}_1 - M_1, V\right) \right\|_2 + \sigma^2 \left\| T\left(\hat{M}_1 - M_1, V\right) \right\|_2 + |\hat{\sigma}^2 - \sigma^2| \left\| T\left(M_1, V\right) \right\|_2\right). \tag{85}
$$

By combining all the inequalities, we get the bound for $S_3$. The bound for $S_4$ can be derived by similar procedure. ∎

To complete the bounds, we need to examine the bounds for the whitening matrix and also the whitened tensors.

### D.3 Properties with whitening matrix

Note that in Algorithm 1 we have $W_3 := T(S_3, W, W, W)$, $W_4 := T(S_4, W, W, W, W)$. To bound $\|W_3\|$ and $\|W_4\|$, we use the fact stated in Section C that these tensor are diagonalized so that finding the norm is actually equivalent to finding the largest eigenvalue of $T(S_3, W, W, W)$ and $T(S_4, W, W, W, W)$, respectively. Note that in Algorithm 1, the first $K_1$ eigenvectors and their corresponding eigenvalues are solved by conducting tensor decomposition on $W_3$, while the others are extracted from $W_4$. With Equation (53) and (57),

$$
\lambda_i = \begin{cases} \frac{-2\pi_i + 1}{\sqrt{\pi_i - \pi_i^2}} & \text{if } i \leq K_1 \\ \frac{6\pi_i^2 - 6\pi_i + 1}{\pi_i - \pi_i^2} & \text{otherwise.} \end{cases} \tag{86}
$$

As we have mentioned previously, eigenvalues of $S_3$ degenerate to zero at the value of $\pi_i = 0.5$ while eigenvalues of $S_4$ degenerate to zero at the value of $\pi_i \approx 0.2, 0.8$. So here we define thresholds, $\pi_{Th_{up}}$ and $\pi_{Th_{down}}$, such that

$$
\frac{-2\pi_{Th_{down}} + 1}{\sqrt{\pi_{Th_{down}} - \pi_{Th_{down}}^2}} = 1, \quad \frac{-2\pi_{Th_{up}} + 1}{\sqrt{\pi_{Th_{up}} - \pi_{Th_{up}}^2}} = -1. \tag{87}
$$

In other words, we solve the latent factors by the third-order moments if $\pi_i < \pi_{Th_{down}}$ or $\pi_i > \pi_{Th_{up}}$, otherwise we turn to the fourth-order moments. Since $\lambda_i$ is a symmetric function of $\pi_i$ on the $\pi_i = 0.5$ axis for $i \in [K]$, we set $\pi_{Th} = \pi_{Th_{down}}$ to simplify the proof. Here we have

$$1 = \left| \frac{-2\pi_{Th} + 1}{\sqrt{\pi_{Th} - \pi_{Th}^2}} \right| \leq |\lambda_i| \leq \frac{-2\pi_{min} + 1}{\sqrt{\pi_{min} - \pi_{min}^2}} \quad \text{if } i \leq K_1 \tag{88}$$

$$-2 \leq \lambda_i \leq \frac{6\pi_{Th}^2 - 6\pi_{Th} + 1}{\pi_{Th} - \pi_{Th}^2} \approx -1 \quad \text{otherwise,} \tag{89}$$

where $\pi_{min} = \text{argmax}_{i \in [K_1]} |\pi_i - 0.5|$. Since $W_3$ and $W_4$ are diagonalized tensor, we have that

$$\|W_3\|_2 \leq \frac{-2\pi_{min} + 1}{\sqrt{\pi_{min} - \pi_{min}^2}}, \quad \|W_4\|_2 \leq 2. \tag{90}$$

Next, in order to bound $\left[ \hat{W}_i - W_i \right]$, we need to consider the bounds using empirical whitening matrix. Let $\hat{W}$ denotes the empirical whitening matrix in our algorithm. Here we define $W := \hat{W}(\hat{W}S_2\hat{W})^{-\frac{1}{2}}$ and $\epsilon_{S_2} := \left\| \hat{S}_{2,k} - S_2 \right\|_2 / \varsigma_k[S_2]$ in order to use the bounds for whitening matrix stated in lemma 10 in [23].

**Lemma 9** *(Lemma 10 in [23]) Assume $\epsilon_{S_2} \leq 1/3$. We have*

1. $W^\top S_2 W = I$, 2. $\left\| \hat{W} \right\|_2 \leq \frac{1}{\sqrt{(1 - \epsilon_{S_2})\varsigma[S_2]}}$,

3. $\left\| \left( \hat{W}S_2\hat{W} \right)^{1/2} - I \right\|_2 \leq 1.5\epsilon_{S_2}, \quad \left\| \left( \hat{W}S_2\hat{W} \right)^{-1/2} - I \right\|_2 \leq 1.5\epsilon_{S_2}$

$$\left\| (\hat{W})^\top A \text{diag}(\pi - \pi^2)^{1/2} \right\|_2 \leq \sqrt{1 + 1.5\epsilon_{S_2}},$$

$$\left\| (\hat{W} - W)^\top A \text{diag}(\pi - \pi^2)^{1/2} \right\|_2 \leq \sqrt{1 + 1.5\epsilon_{S_2}}.$$

Using Lemma 9, we can complete the bounds for empirical whitened tensors.

**Lemma 10** *Assume $\epsilon_{S_2} \leq 1/3$. Then*

$$\left\| \hat{W}_3 - W_3 \right\|_2 \leq \left\| T\left( S_3 - \hat{S}_3, \hat{W}, \hat{W}, \hat{W} \right) \right\|_2 + 3\frac{-2\pi_{min} + 1}{\sqrt{\pi_{min} - \pi_{min}^2}}$$

$$\left\| \hat{W}_4 - W_4 \right\|_2 \leq \left\| T\left( S_4 - \hat{S}_4, \hat{W}, \hat{W}, \hat{W}, \hat{W} \right) \right\|_2 + 10$$

**Proof** Here we only show the second inequality, the first one can be derived with similar procedure.

$$\left\| \hat{W}_4 - W_4 \right\|_2 = \left\| T\left( S_4, W, W, W, W \right) - T\left( \hat{S}_4, \hat{W}, \hat{W}, \hat{W}, \hat{W} \right) \right\|_2$$

$$\leq \left\| T\left( S_4, W, W, W, W \right) - T\left( S_4, \hat{W}, \hat{W}, \hat{W}, \hat{W} \right) \right\|_2 + \left\| T\left( S_4 - \hat{S}_4, \hat{W}, \hat{W}, \hat{W}, \hat{W} \right) \right\|_2 \tag{91}$$

For the first term, using Lemma 9 and Equation (90), we have:

$$\left\| T\left( S_4, W, W, W, W \right) - T\left( S_4, \hat{W}, \hat{W}, \hat{W}, \hat{W} \right) \right\|_2$$

$$\leq \left\| T\left( S_4, \hat{W} - W, \hat{W}, \hat{W}, \hat{W} \right) \right\|_2 + \left\| T\left( S_4, W, \hat{W} - W, \hat{W}, \hat{W} \right) \right\|_2$$

$$+ \left\| T\left( S_4, W, W, \hat{W} - W, \hat{W} \right) \right\|_2 + \left\| T\left( S_4, W, W, W, \hat{W} - W \right) \right\|_2 \tag{92}$$

$$\leq \|T(S_4, W, W, W, W)\|_2 \left\| (\hat{W}^\top S_2 \hat{W})^{1/2} - \mathbf{1} \right\| \left( \left\| (\hat{W}^\top S_2 \hat{W})^{1/2} \right\|_2^3 + \cdots + \left\| (\hat{W}^\top S_2 \hat{W})^{1/2} \right\|_2^0 \right)$$

$$\leq \|T(S_4, W, W, W, W)\|_2 \cdot (1.5\epsilon_{S_2}) \left( (1 + 1.5\epsilon_{S_2})^3 + \cdots (1 + 1.5\epsilon_{S_2}) + 1 \right) \leq 5 \cdot 2 = 10 \tag{93}$$

∎

### D.4 Reconstruction analysis

Before putting everything together, we utilize the eigendecomposition analysis in Appendix C.7 of [23]. First, we consider the case where $A_i$ is recovered by applying tensor decomposition on $W_3$, i.e., for $i \leq K_1$. Note that, in Algorithm 1, $Z_i = \sqrt{\pi_i - \pi_i^2}$. Following the approached in [23], define

$$\gamma_{S_3} := \frac{1}{2 \max_{i \in [K_1]} \sqrt{(\pi_i - \pi_i^2)} \sqrt{eK} \binom{K+1}{2}}, \quad \epsilon_{S_3} := \frac{\left\| W_3 - \hat{W}_3 \right\|}{\gamma_{S_3}},$$

$$\gamma_{S_4} := \frac{1}{2 \max_{i > K_1} \sqrt{(\pi_i - \pi_i^2)} \sqrt{eK} \binom{K+1}{2}}, \quad \epsilon_{S_4} := \frac{\left\| W_4 - \hat{W}_4 \right\|}{\gamma_{S_4}}.$$

**Lemma 11** *(Reconstruction Accuracy in [23]) Assume $\epsilon_{S_2} \leq 1/3$, $\epsilon_{S_3} \leq 1/4$ and $\epsilon_{S_4} \leq 1/4$, and $\epsilon_1 \leq 1/3$. With $\kappa [S_2] := \varsigma_1 [S_2] / \varsigma_K [S_2]$, there exists a permutation $\pi$ on $[K]$ such that*

$$\left\| A_{\pi(i)} - \hat{A}_i \right\| \leq 3 \left\| A_{\pi(i)} \right\|_2 \epsilon_{1,i} + 2 \|S_2\|_2^{1/2} \epsilon_{0,i}, \quad \forall i \in [K]$$

*where*

$$\epsilon_{0,i} = \begin{cases} (5.5\epsilon_{S_2} + 7\epsilon_{S_3}) / \sqrt{\pi_{min} - \pi_{min}^2} & \text{if } i \in [K_1] \\ 13.75\epsilon_{S_2} + 17.5\epsilon_{S_4} & \text{otherwise} \end{cases},$$

$$\epsilon_{1,i} = \begin{cases} \left( \left( 6.875\kappa [S_2]^{1/2} + 2 \right) \epsilon_{S_2} + \left( 8.75\kappa [S_2]^{1/2} + \gamma_{S_3} \sqrt{\pi_{min} - \pi_{min}^2} \right) \epsilon_{S_3} \right) \\ \qquad\qquad / \left( \gamma_{S_3} \sqrt{\pi_{min} - \pi_{min}^2} \right) & \text{if } i \in [K_1] \\ 2.5 \left( \left( 6.875\kappa [S_2]^{1/2} + 2 \right) \epsilon_{S_2} + \left( 8.75\kappa [S_2]^{1/2} + 0.4\gamma_{S_4} \right) \epsilon_{S_4} \right) / \gamma_{S_4} & \text{otherwise} \end{cases}$$

### D.5 Proof of Theorem 3

The proof can be accomplished by finding out the sufficient sample size that satisfies the assumption in Lemma 11. First, assume that $n \geq Ck \log(1/\delta)$ to make sure $\varepsilon_w \leq 1$. In this proof, we use $c, c_1, c_2, \cdots$ to denote some positive constant. By Lemma 5 and 6, with probability greater than $1 - \delta$, bounds for first and second -order moments are

$$\left\| \hat{M}_1 - M_1 \right\|_2 \leq c_1 \sigma \sqrt{\frac{d + \log(2^k/\delta)}{\tilde{\pi}n}} + c_1 \sum_{i=1}^{K} \|A_i\|_2 \sqrt{\frac{2^K \log(1/\delta)}{n}} \tag{94}$$

$$\left\| \hat{M}_2 - M_2 \right\|_2 \leq c_1 \left( \sigma^2 \sqrt{\frac{d + \log(2^k/\delta)}{\tilde{\pi}n}} + \sigma^2 \frac{d + \log(2^k/\delta)}{\tilde{\pi}n} + \sigma \sqrt{\frac{d + \log(2^k/\delta)}{\tilde{\pi}n}} \right) \tag{95}$$

$$+ c_1 \left( \sum_{i=1}^{K} \|A_i\|_2^2 + \sigma^2 \right) \sqrt{\frac{2^K \log(1/\delta)}{n}} \tag{96}$$

$$\leq c_1 \left( 2 \left( \sum_{i=1}^{K} \|A_i\|_2^2 + \sigma^2 \right) \sqrt{\frac{d + \log(2^k/\delta)}{\tilde{\pi}n}} + \sigma^2 \frac{d + \log(2^k/\delta)}{\tilde{\pi}n} \right) \tag{97}$$

Using Lemma 7, we derive the bound for $\sigma$ and approximate rank-k second-order tensor as

$$\max \left\{ \left| \hat{\sigma}^2 - \sigma^2 \right|, \left\| \hat{S}_{2,K} - S_2 \right\|_2 \right\}$$

$$\leq 4c_1 \left( 2 \left( \sum_{i=1}^{K} \|A_i\|_2^2 + \sigma^2 \right) \sqrt{\frac{d + \log(2^k/\delta)}{\tilde{\pi}n}} + \sigma^2 \frac{d + \log(2^k/\delta)}{\tilde{\pi}n} \right)$$

$$+ 8c_1^2 \left( \sigma \sqrt{\frac{d + \log(2^k/\delta)}{\tilde{\pi}n}} + \sum_{i=1}^{K} \|A_i\|_2 \sqrt{\frac{2^K \log(1/\delta)}{n}} \right)^2$$

$$+ 8c_1 \|M_1\|_2 \left( \sigma \sqrt{\frac{d + \log(2^k/\delta)}{\tilde{\pi}n}} + \sum_{i=1}^{K} \|A_i\|_2 \sqrt{\frac{2^K \log(1/\delta)}{n}} \right) \tag{98}$$

$$\leq c_2 \left( \sum_{i=1}^{K} \|A_i\|_2^2 + \sigma^2 \right) \left( \sqrt{\frac{d + \log(2^k/\delta)}{\tilde{\pi}n}} + \frac{d + \log(2^k/\delta)}{\tilde{\pi}n} \right). \tag{99}$$

To ensure that

$$\max\left\{\frac{|\hat{\sigma}^2 - \sigma^2|}{\varsigma_K\left(S_2\right)}, \epsilon_{S_2}\right\} \leq c_3 \frac{\gamma_{S_3}^2 \tilde{\pi}}{\kappa\left[S_2\right]^{1/2}} \leq 1/3, \tag{100}$$

we set the sample size as

$$n \geq c\frac{d + \log\left(2^k/\delta\right)}{\tilde{\pi}}\left(\left[\frac{\kappa\left[S_2\right]^{1/2}\left(\sum\limits_{i=1}^{K}\|A_i\|_2^2 + \sigma^2\right)}{\gamma_{S_3}^2 \tilde{\pi}\varsigma_K\left[S_2\right]\epsilon}\right]^2 + \left[\frac{\kappa\left[S_2\right]^{1/2}\left(\sum\limits_{i=1}^{K}\|A_i\|_2^2 + \sigma^2\right)}{\gamma_{S_3}^2 \tilde{\pi}\varsigma_K\left[S_2\right]\epsilon}\right]\right).$$

To examine the moments after multiplying whitening matrix $W$, by Lemma 9,

$$\left\|\hat{W}\right\|_2 \leq \sqrt{1.5/\varsigma_K\left[S_2\right]} \tag{101}$$

$$\max_{z_i \in [2^K]}\left\|T\left(M_{1,z_i}, \hat{W}\right)\right\| \leq \left\|\hat{W}^\top A \mathrm{diag}\left(\pi - \pi^2\right)^{1/2}\right\|_2 / \sqrt{\pi_{min} - \pi_{min}^2} \tag{102}$$

$$\leq \sqrt{1.5/\left(\pi_{min} - \pi_{min}^2\right)} \tag{103}$$

$$\max_{z_i \in [2^K]}\left\|T\left(M_{2,z_i}, \hat{W}, \hat{W}\right)\right\| \leq 1.5/\left(\pi_{min} - \pi_{min}^2\right) + \sigma^2\left(1.5/\varsigma_K\left[S_2\right]\right) \tag{104}$$

$$\max_{z_i \in [2^K]}\left\|T\left(M_{3,z_i}, \hat{W}, \hat{W}, \hat{W}\right)\right\| \leq \left(1.5/\left(\pi_{min} - \pi_{min}^2\right)\right)^{3/2}$$
$$+ 3\sigma^2\sqrt{1.5/\left(\pi_{min} - \pi_{min}^2\right)}\left(1.5/\varsigma_K\left[S_2\right]\right) \tag{105}$$

$$\max_{z_i \in [2^K]}\left\|T\left(M_{4,z_i}, \hat{W}, \hat{W}, \hat{W}, \hat{W}\right)\right\| \leq \left(1.5/\left(\pi_{min} - \pi_{min}^2\right)\right)^2 + 6\sigma^2\frac{2.25}{\left(\pi_{min} - \pi_{min}^2\right)\varsigma_K\left[S_2\right]}$$
$$+ 3\sigma^4\left(1.5/\varsigma_K\left[S_2\right]\right)^2 \tag{106}$$

Using Lemma 6,

$$\left\|T\left(\hat{M}_1 - M_1, \hat{W}\right)\right\| \leq c_4\frac{\sigma}{\varsigma_K\left[S_2\right]^{1/2}}\sqrt{\frac{K + \log\left(2^K/\delta\right)}{\tilde{\pi}n}} + c_4\frac{1}{\sqrt{\pi_{min} - \pi_{min}^2}}\sqrt{\frac{2^K\log\left(1/\delta\right)}{n}} \tag{107}$$

$$\left\|T\left(\hat{M}_2 - M_2, \hat{W}, \hat{W}\right)\right\| \leq c_4\frac{\sigma^2}{\varsigma_K\left[S_2\right]}\left(\sqrt{\frac{K + \log\left(2^K/\delta\right)}{\tilde{\pi}n}} + \frac{K + \log\left(2^K/\delta\right)}{\tilde{\pi}n}\right)$$
$$+ c_4\frac{1}{\sqrt{\pi_{min} - \pi_{min}^2}\varsigma_K\left[S_2\right]^{1/2}}\sqrt{\frac{2^K\log\left(1/\delta\right)}{n}}$$
$$+ c_4\left(\frac{1}{\varsigma_K\left[S_2\right]} + \frac{1}{\sqrt{\pi_{min} - \pi_{min}^2}}\right)\sqrt{\frac{K\log\left(1/\delta\right)}{n}} \tag{108}$$

$$\left\|T\left(\hat{M}_3 - M_3, \hat{W}, \hat{W}, \hat{W}\right)\right\| \leq c_4\frac{\sigma^3}{\varsigma_K\left[S_2\right]^{3/2}}\sqrt{\frac{\left(K + \log\left(2^K/\delta\right)\right)^3}{\tilde{\pi}n}}$$
$$+ c_4\frac{\sigma^2}{\varsigma_K\left[S_2\right]\sqrt{\pi_{min} - \pi_{min}^2}}\left(\sqrt{\frac{K + \log\left(2^K/\delta\right)}{\tilde{\pi}n}} + \frac{K + \log\left(2^K/\delta\right)}{\tilde{\pi}n}\right)$$
$$+ c_4\frac{\sigma}{\varsigma_K\left[S_2\right]^{1/2}\left(\pi_{min} - \pi_{min}^2\right)}\sqrt{\frac{K + \log\left(2^K/\delta\right)}{\tilde{\pi}n}}$$
$$+ c_4\left(\frac{\sigma^2}{\varsigma_K\left[S_2\right]\sqrt{\pi_{min} - \pi_{min}^2}} + \frac{1}{\left(\pi_{min} - \pi_{min}^2\right)^{3/2}}\right)\sqrt{\frac{K\log\left(1/\delta\right)}{n}} \tag{109}$$

$$\left\| T\left( \hat{M}_4 - M_4, , \hat{W}, \hat{W}, \hat{W}, \hat{W} \right) \right\| \leq c_4 \frac{\sigma^4}{\varsigma_K\left[S_2\right]^2} \left( \left( \frac{K + \log\left(2^K/\delta\right)}{\tilde{\pi}n} \right) + \left( \frac{K + \log\left(2^K/\delta\right)}{\tilde{\pi}n} \right)^{3/2} \right)$$

$$+ c_4 \frac{\sigma^3}{\varsigma_K\left[S_2\right]^{3/2}} \sqrt{\frac{\left(K + \log\left(2^K/\delta\right)\right)^3}{\tilde{\pi}n}}$$

$$+ c_4 \frac{\sigma^2}{\varsigma_K\left[S_2\right]} \left( \sqrt{\frac{K + \log\left(2^K/\delta\right)}{\tilde{\pi}n}} + \frac{K + \log\left(2^K/\delta\right)}{\tilde{\pi}n} \right)$$

$$+ c_4 \frac{\sigma}{\varsigma_K\left[S_2\right]^{1/2}} \sqrt{\frac{K + \log\left(2^K/\delta\right)}{\tilde{\pi}n}}$$

$$+ c_4 \left( \frac{\sigma^4}{\varsigma_K\left[S_2\right]} + \frac{\sigma^2}{\varsigma_K\left[S_2\right]} \right) \sqrt{\frac{K \log\left(1/\delta\right)}{n}}. \tag{110}$$

With Lemma 8 and 10,

$$\left\| T\left( \hat{S}_2 - S_2, \hat{W}, \hat{W} \right) \right\|_2 \leq \left\| T\left( \hat{M}_2 - M_2, \hat{W}, \hat{W} \right) \right\|_2 + \left\| T\left( \hat{M}_1 - M_1, \hat{W} \right) \right\|_2^2$$

$$+ 2 \frac{\pi_{max}}{\sqrt{\pi_{max} - \pi_{max}^2}} \left\| T\left( \hat{M}_1 - M_1, \hat{W} \right) \right\|_2 + \frac{1.5}{\varsigma_K\left[S_2\right]} \left| \hat{\sigma}^2 - \sigma^2 \right| \tag{111}$$

$$\left\| T\left( \hat{S}_3 - S_3, \hat{W}, \hat{W}, \hat{W} \right) \right\| \leq \left\| T\left( \hat{M}_3 - M_3, \hat{W}, \hat{W}, \hat{W} \right) \right\|_2$$

$$+ \left( \left\| T\left( \hat{M}_1 - M_1, \hat{W} \right) \right\|_2 + \frac{\pi_{max}}{\sqrt{\pi_{max} - \pi_{max}^2}} \right)^3 - \left( \frac{\pi_{max}}{\sqrt{\pi_{max} - \pi_{max}^2}} \right)^3$$

$$+ 3 \left( \left\| T\left( \hat{M}_1 - M_1, \hat{W} \right) \right\|_2 \left\| T\left( \hat{S}_2 - S_2, \hat{W}, \hat{W} \right) \right\|_2 + \frac{\pi_{max}}{\sqrt{\pi_{max} - \pi_{max}^2}} \left\| T\left( \hat{S}_2 - S_2, \hat{W}, \hat{W} \right) \right\|_2 \right.$$

$$+ \left\| T\left( \hat{M}_1 - M_1, \hat{W} \right) \right\|_2 \frac{-2\pi_{min} + 1}{\sqrt{\pi_{min} - \pi_{min}^2}} \right) + \frac{4.5}{\varsigma_K\left[S_2\right]} \left( \left| \hat{\sigma}^2 - \sigma^2 \right| \left\| T\left( \hat{M}_1 - M_1, \hat{W} \right) \right\|_2 \right.$$

$$+ \sigma^2 \left\| T\left( \hat{M}_1 - M_1, \hat{W} \right) \right\|_2 + \left| \hat{\sigma}^2 - \sigma^2 \right| \frac{\pi_{max}}{\sqrt{\pi_{max} - \pi_{max}^2}} \right). \tag{112}$$

Plug this in Lemma 10, we get the overall bounds for $\left\| W_3 - \hat{W}_3 \right\|$. To get $\epsilon_{S_3} \leq c_5 \frac{\gamma_{S_3}\sqrt{\tilde{\pi}}}{\kappa[S_2]^{1/2}} \epsilon$, we set

$$n \geq \text{poly}\left( d, K, \frac{1}{\epsilon}, \log(1/\delta), \frac{1}{\tilde{\pi}}, \frac{\varsigma_1\left[S_2\right]}{\varsigma_K\left[S_2\right]}, \frac{\sum_{i=1}^{K} \|A_i\|_2^2}{\varsigma_K\left[S_2\right]}, \frac{\sigma^2}{\varsigma_K\left[S_2\right]}, \frac{1}{\sqrt{\pi_{min} - \pi_{min}^2}}, \frac{\pi_{max}}{\sqrt{\pi_{max} - \pi_{max}^2}} \right) \tag{113}$$

Similarly, for $A_i$ reconstructed by $\hat{W}_4$, n should be set to

$$n \geq \text{poly}\left( d, K, \frac{1}{\epsilon}, \log(1/\delta), \frac{1}{\tilde{\pi}}, \frac{\varsigma_1\left[S_2\right]}{\varsigma_K\left[S_2\right]}, \frac{\sum_{i=1}^{K} \|A_i\|_2^2}{\varsigma_K\left[S_2\right]}, \frac{\sigma^2}{\varsigma_K\left[S_2\right]} \right), \tag{114}$$

in order to $\epsilon_{S_4} \leq c_6 \frac{\gamma_{S_4}\sqrt{\tilde{\pi}}}{\kappa[S_2]^{1/2}} \epsilon$. The overall bounds can be obtained by Equation 113, 114 and Lemma 11.

# E  Tail Inequalities

Here we derive the tail inequality for the fourth-order subgaussian random tensor.

**Lemma 12** *Let $x_1, x_2, \cdots, x_n$ be i.i.d. random variables such that*

$$\mathbf{E}_i\left[\exp\left(\eta x_i\right)\right] \leq \exp\left(\gamma\eta^2/2\right) \quad \forall \eta \in \mathbb{R} \tag{115}$$

*Then for any $t > 0$ and $\frac{\gamma t}{n} < \frac{1}{4}$,*

$$\Pr\left[\frac{1}{n}\sum_{i=1}^{n}\left(x_i^4 - \mathbf{E}_i\left[x_i^4\right]\right) > \gamma\sqrt{\frac{64t^2}{n^2}\left(8\gamma - \frac{16\gamma^2 t}{n}\right)\frac{1}{\left(1 - 4\gamma\frac{t}{n}\right)^2}}\right] \le e^{-t}, \tag{116}$$

$$\Pr\left[\frac{1}{n}\sum_{i=1}^{n}\left(x_i^4 - \mathbf{E}_i\left[x_i^4\right]\right) < -\gamma\sqrt{\frac{8t^2}{n^2}\left(2\gamma + \frac{\gamma^2 t}{n}\right)\frac{1}{\left(1 + \gamma\frac{t}{n}\right)^2}}\right] \le e^{-t}, \tag{117}$$

**Proof** We use Chernoff's bounding method to derive the inequality. For $\eta < \frac{1}{2\epsilon\gamma}$, set $\eta = \frac{1-\sigma}{2\gamma\epsilon}$ for some $\sigma > 0$, we have

$$\mathbf{E}_i\left[\exp(\eta x_i^4)\right] = 1 + \eta\mathbf{E}_i\left[x_i^4\right] + \eta\int_0^\infty \left(\exp\left(\eta\epsilon^2\right) - 1\right)\mathbf{E}_i\left[\mathbb{1}_{\{x_i^4 > \epsilon^2\}}\right]d\epsilon^2 \tag{118}$$

$$\le 1 + \eta\mathbf{E}_i\left[x_i^4\right] + 2\eta\int_0^\infty \left(\exp\left(\eta\epsilon^2\right) - 1\right)\exp\left(\frac{-\epsilon}{2\gamma}\right)2\epsilon d\epsilon \tag{119}$$

$$\le 1 + \eta\mathbf{E}_i\left[x_i^4\right] + 4\eta\left(\int_0^\infty \epsilon\exp\left(\frac{-\sigma\epsilon}{2\gamma}\right)d\epsilon - \int_0^\infty \epsilon\exp\left(\frac{-\epsilon}{2\gamma}\right)d\epsilon\right) \tag{120}$$

$$\le 1 + \eta\mathbf{E}_i\left[x_i^4\right] + 4\eta\left(4\gamma^2\left(\frac{1}{\sigma^2} - 1\right)\right) \tag{121}$$

$$\le \exp\left(\eta\mathbf{E}_i\left[x_i^4\right] + 4\eta\left(4\gamma^2\left(\frac{1}{\sigma^2} - 1\right)\right)\right) \tag{122}$$

The second line uses the fact that $\Pr\left[x_i^4 > \epsilon^2\right] \le \frac{\mathbf{E}[\exp(\alpha|x_i|)]}{\exp(\alpha\epsilon^{1/2})} \le 2\frac{\exp(\gamma\alpha^2/2)}{\exp(\alpha\epsilon^{1/2})} = 2\exp\left(-\frac{\epsilon}{2\gamma}\right)$ with $\alpha = \frac{\epsilon^{1/2}}{\gamma}$. Since the above inequality holds for $i = 1, 2, \cdots, n$,

$$\mathbf{E}\left[\exp\left(\eta\sum_{i=1}^{n}\left(x_i^4 - \mathbf{E}_i\left[x_i^4\right]\right)\right)\right] = \prod_{i=1}^{n}\mathbf{E}_i\left[\exp\left(\eta\left(x_i^4 - \mathbf{E}_i\left[x_i^4\right]\right)\right)\right] \tag{123}$$

$$\le \exp\left(16n\eta\gamma^2\left(\frac{1}{\sigma^2} - 1\right)\right) \tag{124}$$

With Chernoff's inequality, for $0 \le \eta < \frac{1}{2\epsilon\gamma}$ and $\epsilon \ge 0$,

$$\Pr\left[\frac{1}{n}\sum_{i=1}^{n}\left(x_i^4 - \mathbf{E}_i\left[x_i^4\right]\right) > \epsilon\right] \le \exp\left(-\eta n\epsilon + 16n\eta\gamma^2\left(\frac{1}{\sigma^2} - 1\right)\right). \tag{125}$$

Setting $\eta = \frac{1-\sigma}{2\gamma\epsilon}$ and $\sigma = 1 - \frac{4\gamma t}{n}$, for $\frac{\gamma t}{n} < \frac{1}{4}$, we get the first inequality. For $\eta < 0$ and $\epsilon \ge 0$,

$$\Pr\left[\frac{1}{n}\sum_{i=1}^{n}\left(x_i^4 - \mathbf{E}_i\left[x_i^4\right]\right) < -\epsilon\right] \le \exp\left(\eta n\epsilon + 16n\eta\gamma^2\left(\frac{1}{\sigma^2} - 1\right)\right). \tag{126}$$

Setting $\sigma = 1 + \gamma t$ gives the claimed inequality. ∎

**Lemma 13** *(Fourth-order normal random vectors). Let $y_1, y_2, \cdots y_n \in \mathbb{R}^d$ be i.i.d. $N(0, I)$ random vectors. For $\epsilon_0 \in (0, 1/4)$ and $\delta \in (0, 1)$,*

$$\Pr\left[\left\|\frac{1}{n}\sum_{i=1}^{n}y_i \otimes y_i \otimes y_i \otimes y_i - \mathbf{E}\left[\epsilon \otimes \epsilon \otimes \epsilon \otimes \epsilon\right]\right\|_2 > \frac{1}{1 - 4\epsilon_0}\epsilon_{\epsilon_0, t, n}\right] \le 2\delta \tag{127}$$

*where*

$$\epsilon_{\epsilon_0, t, n} = \sqrt{\frac{2048\ln\left((1 + 2/\epsilon_0)^d/\delta\right)^2}{n^2} + \frac{8\ln\left((1 + 2/\epsilon_0)^d/\delta\right)^3}{n^3}} \tag{128}$$

**Proof** We follow the approach of [22]. Let $Y := \frac{1}{n}\sum_{i=1}^{n}y_i \otimes y_i \otimes y_i \otimes y_i - \mathbf{E}\left[\epsilon\mathbf{1} \otimes \epsilon\mathbf{1} \otimes \epsilon\mathbf{1} \otimes \epsilon\mathbf{1}\right]$. By [28], there exists $Q \subseteq \mathcal{S}^{d-1} := \{\alpha \in \mathbb{R}^d : \|\alpha\|_2 = 1\}$ with cardinality at most $(1 + 2\epsilon)^d$ such that $\forall \alpha \in \mathcal{S}^{d-1} \exists q \in Q$ $\|\alpha - q\|_2 \le \epsilon_0$. Since, for any $q \in Q$, $y_i^\top q$ is distributed as $N(0, 1)$, with union bounds and Lemma 12, for

$\Pr\left[\exists q \in Q \left|T\left(Y, q, q, q, q\right)\right| > \epsilon_{\epsilon_0, t, n}\right] \leq 2\delta$. So we assume with probability greater than $1 - 2\delta$, $\forall q \in Q$, $\left|T\left(Y, q, q, q, q\right)\right| \leq \epsilon_{\epsilon_0, t, n}$. Let $\alpha_0 = \operatorname{argmax}_{\alpha \in \mathcal{S}^{d-1}} \left|T\left(Y, \alpha, \alpha, \alpha, \alpha\right)\right|$, we have

$$\|Y\|_2 = \left|T\left(Y, \alpha_0, \alpha_0, \alpha_0, \alpha_0\right)\right| \tag{129}$$

$$\leq \min_{q \in Q} \left|T\left(Y, q, q, q, q\right)\right| + \left|T\left(Y, \alpha_0 - q, q, q, q\right)\right| + \left|T\left(Y, \alpha_0, \alpha_0 - q, q, q\right)\right|$$

$$+ \left|T\left(Y, \alpha_0, \alpha_0, \alpha_0 - q, q\right)\right| + \left|T\left(Y, \alpha_0, \alpha_0, \alpha_0, \alpha_0 - q\right)\right| \tag{130}$$

$$\leq \min_{q \in Q} \left|T\left(Y, q, q, q, q\right)\right| + 4 \|\alpha_0 - q\| \|Y\|_2 \tag{131}$$

$$\leq \epsilon_{\epsilon_0, t, n} + 4\epsilon_0 \|Y\|_2 , \tag{132}$$

which yields

$$\|Y\|_2 \leq \frac{1}{1 - 4\epsilon_0} \cdot \epsilon_{\epsilon_0, t, n} \tag{133}$$

$\blacksquare$