[Reviews · NeurIPS 2014]

Submitted by Assigned_Reviewer_19

Spectral methods are based on decomposing moment tensors. If data are generated from latent variable models, their empirical moment tensors have a kind of (approximate) low-rank decomposition (approximate due to both noisy observations and finite sample estimation error in the empirical moments). These decompositions can be computed from the moment tensors, e.g. using a kind of power iteration method on related (symmetrized) tensors. The basic ideas of these spectral methods are fairly well established and many examples have been explored, especially in [6].

This paper applies spectral fitting methods to data modeled as being generated from an IBP, including two common emission models (a linear gaussian model and a sparse factor analysis model, described in Section 2). The main ingredients are a calculation of the appropriate moment tensors and corresponding symmetrized versions (Section 3), an application of the standard tensor power decomposition method (Section 4), and concentration proofs to offer recovery guarantees when data are generated from the model (Section 5). There are also experiments (Section 6).

The paper applies existing methodology and 'turns the crank' in the context of the IBP. While having such methods available for the IBP is interesting to the community, their development would fit better as an example in a paper like [6] than a full NIPS paper. Therefore the technical contribution and originality seem low.

There could be contributions here that I'm not appreciating. For example, there could be substantial deviations from the standard machinery that are required, either for the algorithm or for the analysis in Section 5. In developing the algorithm, there is a need to use fourth-order moments (Section 3.1), though this doesn't seem like a significant leap. In the analysis of Section 5 the authors suggest (around line 281) that they provide a more general methodology for producing relevant concentration of measure bounds, but again it is not made clear whether this is a contribution of great originality or significance outside of the scope of this paper. If there are contributions in this paper that differentiate it from a 'turn the crank' application of standard machinery, the authors should highlight them in their response and clarify them in the text.

The paper is decently written overall, though some points that seem relevant are not discussed. For example, Section 5 seems cut short and seems a bit disconnected from the rest of the paper. It could benefit substantially from more explicit connections back to the IBP. In addition, it could be made clear how the notion of concentration of measure interacts with the nonparametric generative model; does the size of K grow with the number of samples m? (The references [6,7] do not mention the Dirichlet process explicitly as far as I can tell, so I'm not sure if the nonparametric aspect of the story is made clear in some existing work without a more direct reference.) Also, does the notation for the number of samples switch between m and n in Sections 5.1 and 5.2?

The experiments in Section 6 seem a bit weak. Both the first and second involve data that are essentially generated from the model (the first being synthetic data and the second being the IBP classic of objects-in-images) and therefore they don't shed much light on whether the method can perform well when data are not extremely close to following the model's assumptions, a question which seems especially relevant for spectral methods. In the short discussion of the third experiment, there isn't much to reveal the advantages and disadvantages of the proposed spectral algorithm over other algorithms. If the proposed method and analysis are essentially an application of existing machinery to the IBP, then experiments could provide a real contribution in shedding light on when spectral methods may be appropriate and when they may fail. But this experiments section does not provide such insight.

Here are some miscellaneous points:
- Since [6,7] don't mention the Dirichlet process explicitly (based on searching the text), the reference around line 38 seems strange.
- Line 113 says all proofs are deferred to the appendix, but one proof remains in the paper.
- The {0;1} on line 134 should probably be {0,1}, though that sentence could be revised for clarity.
- The comment on line 142 that "[t]his is undesirable for the power method" could be clarified.
- The plots in Figure 1 look like they were saved to a raster format, which makes them hard to read. The text and line sizes should probably be adjusted.
- In Figure 3, was the MCMC algorithm stopped short at 1.72 seconds, or was there some reason to believe it was stuck in such a suboptimal mode?
Summary: While the topic is interesting, the paper explains an essentially standard application of spectral methods to IBP models and does not shed light on when to choose the spectral method over more traditional ones.

Submitted by Assigned_Reviewer_21

Spectral methods have recently been proposed as an alternative and efficient framework for inference based on characterizing moments in order to identify model parameters. In this paper the authors extend this framework to the Indian Buffet Process (IBP) as well as two models based on the IBP namely the linear Gaussian Latent Factor Model and the Infinite Sparse Factor Analysis model based on Laplace and Gaussian priors forming respectively a sparse PCA and ICA type of decomposition. Multilinearity of higher order moments are explored and non-identifiability issues in the third order moment is handled by also analyzing the structure of the fourth order moment. An algorithm is proposed for spectral inference and the framework demonstrated to be more efficient than Variational and MCMC inference. Extensive derivations of moments and proofs of bounds are provided in the accompanying appendix.

The paper builds on the existing ideas of spectral approaches (ref. 5-8) and presently extend these to the Indian Buffet Process. This extension including proofs of bounds is non-trivial and the very condensed paper provides a very technical and detailed mathematical analysis which may warrants its publication. From an application point of view the method appears both to be simple to implement and to work well in practice which may make the framework be adapted by other researchers giving the paper a substantial impact.

While the experiments indicate that the method is more efficient than a variational approach it would improve the paper to analyze how the approach compares in accuracy to a full MCMC procedure. Comparison is provided to MCMC for one synthetic example and it is shown that in less computational time better reconstruction is achieved. However, how would the method compare to the MCMC procedure run for much longer? The results for the real gene expression data also appear to be meaningful but again how do these results compare to Variational and MCMC inference? – and do these existing alternative approaches also identify in the order of 10 (similar) components? Clearly, the spectral approach proposed seems to be accurate and very efficient but it would be interesting to see in practice how accurate the results are compared to more exhaustive MCMC procedures.

In order to determine the dimensionality K in algorithm 1 a truncation to eigenvalues larger than epsilon is applied in order to determine the rank of S_2. As determination of K is very central to non-parametric approaches it would improve the paper to further discuss the influence of truncation for determining the order K and how epsilon/the truncation was chosen. In general the spectral approaches admit to quantify model order by truncating eigenvalue decompositions as also remarked by the authors. Thus it would improve the paper to elaborate on how easily this truncation level can be defined in practice. It is mentioned that K can be determined by the largest slope of the eigenvalue spectrum – how reliable can this largest slope be identified in general?

The orthogonal tensor decomposition requires a number of random initializations to be reliably estimated. How many random initializations are needed in practice and how severely do local optima impact the results? (It is unclear in the paper how critical this issue is.)

Minor comments:
Synthetic dataset figures to the right: should 0101 not be 0110 and 1001 be 1010? Otherwise, please explain how the binary notation is used to reflect the symbols learned.

The notation is not always clear in terms of what are scalars and vectors. I.e., \pi is a vector but when writing \pi^2 I believe element wise exponentiation is used. Some places C_x appears as vectors but other places they are treated as scalars. Please clarify this notation to make the paper more assessable. Also diag is used both some places to form a diagonal matrix but also to form an order fourth diagonal tensor.

Minor comments to the extensive appendix:
Line 511: x^T A -> v^T A

Equation 32 is top -> \top
Equation 36 please clarify how E_x becomes only E_z in line 36 and then in line 37 E_z, E_y.
Line 669: In this section, we provides bounds for moments of linear gaussian latent feature model -> In this section, we provide bounds for moments of the linear gaussian latent feature model

Line 968: Before starting the put everything together -> Before starting to put everything together

Line 1126: in order to – do you here mean: in order for / in order to assure?

Please explain the Poly(.) notation.

Line 1126: Lamma 11 -> Lemma 11
Summary: The authors extend the existing spectral approaches to the Indian Buffet Process including the Linear Gaussian Latent Feature Model and Infinite Sparse Factor Analysis providing what appears to be a useful and very efficient inference framework. Considering non-parametric models it is unclear how well the model orders in general can be determined.

Submitted by Assigned_Reviewer_38

The paper developed spectral algorithms for the models of [18] and [24]
by developing general theory for the IBP and applying it. Algorithm 1
is a precise description of the approach. Very impressive.

Theorem 3, a challenging result, doesn't really imply anything in
practice, other that "eventually it works", so experimentation is
really needed. Can you say anything about the nature of the
polynomial?

The experimental work looks at some toy data from [18]
and then considers one gene expression data problem.
The experimental work is thus suggestive but not
convincing in any way. More comparisons are needed.

Now it seems your experimental work is on par with the
standards of the original papers, such as [24]. Their experimental
work is "illustrative" since they have a new model.
Your experimental work needs to be more thorough in showing the
comparison with the original. You work on a few toy
examples. The interesting result, Fig. 4 is not done for MCMC.
Was it able to cope with the size? Perhaps not.

As an example, consider Reed and Ghahramani ICML 2013.
They give a variety of algorithms compared on
realistic looking data sets, including an analysis
of computational complexity.

Regarding introductory statement:
"the issue of spectral inference in Dirichlet Processes is largely settled [6,7]"
This statement is completely unsupported by the references cited.

Equations (2) and (3): you mean IBP(\alpha) ?
Summary: Impressive spectral algorithm with a convergence result looks good, but the experimental
comparison is on toy data, whereas the standard for comparisons for new algorithms
is set by Reed and Ghahramani 2013 ICML. Better experimental work needed since the
theory is very vague on performance.

Submitted by Assigned_Reviewer_42

This paper shows how to apply a method-of-moments technique to
inference in the Indian Buffet Process (IBP). The IBP is an important
nonparametric Bayesian model and has many applications.
Existing inference methods based on sampling are computationally
expensive. The method-of-moments approach to inference developed in
this paper is based on the tensor decomposition approach from [6]. The
main contributions of the present paper are: the moment analysis for
the IBP that motivate the proposed method, computational and sample
complexity analysis for the proposed inference method, and a
demonstration of the method on some synthetic and image recovery
problems, showing a favorable comparison to the "infinite variational
approach".

The paper quite well-written and is well-motivated. One takeaway from
this paper is the view of nonparametric models as "effectively
parametric" models, which permits a "parameter estimation" approach to
inference. This is quite similar to approach taken in [7] for
inference in the mixed-membership block model. I think this is a nice
idea that is worth promoting, and the present paper does this well.

As mentioned above, the main technical contribution is the moment
analysis for the IBP. One interesting aspect of the IBP is that the
third and fourth order moments can be missing some components with
certain "dish probabilities", but together they contain all of the
components.

The concentration inequalities are fairly standard, although there is
a slight improvement over the loose analysis from [23]. The error
analysis essentially follows [23,8]. Much of the discussion in Section
4 on using random projections and the "projected" tensor power method
in an efficient way is actually from [6] (and should be properly
attributed).

Line 91: "z ~ IBP(z)" --- what does this mean?
Line 223: "(...)_epsilon" --- what's epsilon?
Line 283: "average-median theorem" --- where is this used?
Line 582: "convexity of expectations" --- probably you just mean
Jensen's inequality and convexity of the Euclidean norm?

[I've read the author rebuttal.]
Summary: In summary, the paper is a well-written description of how to use a
method-of-moments based on tensor decompositions for inference in a
particular nonparametric Bayesian model (IBP). There is some
algorithmic and other technical novelty here, though not a whole lot;
still, I think the paper is an interesting contribution and should be
of interest to the machine learning community.
Author Feedback
Author rebuttal: “Technical Novelty”

We agree that our work follows the line of [6]. Alas, an inclusion in the survey is impossible since [6] is written by an entirely different set of authors.

In terms of application, we choose a statistical inference problem that has not received sufficient practical interest due to its computational cost and we provide an efficient spectral algorithm with theoretical guarantees. In this vein, the paper is as relevant as, say, a variational approach or a sampling approach to IBP inference. With the main exception that our proposed algorithm is orders of magnitudes faster and highly accurate.

In terms of technical novelty, besides the the novel statistical model, we address the issue of higher order tensors, and we provide uniform convergence bounds that work directly on the effect of the tensor on test vectors. This allows us to derive bounds for all orders in one single derivation rather than requiring different analysis for each order, at least in the case where moments are bounded.

“Comparison to Variational and MCMC Inference”

For the gene expression dataset we did not show the results derived by variational inference since they did not display any significant attributes on both the genes and their projection values over different kind of treatments (i.e. the results were of poor quality). Besides, CPU cost was high when compared to spectral methods. For MCMC inference we terminated the run after five days since the machine was required for other tasks. We examined these results but again no salient characteristics could be obtained from the solutions. In short, both variational methods and MCMC failed miserably.

However, in order to show that our method has the ability to find meaningful results in real-world data like gene expression data, we present the results in a way to easily follow the intuition in our derivation.

“MCMC run in Figure 3”

MCMC got stuck in a local minimum (of perplexity) at 1.72s. Continuing the run for several minutes after that did not improve performance. In other words, the sampler did not mix well. Possibly a very different sampler other than the one proposed by the inventors of the IBP might mix faster but this is not within the scope of the present paper focused on spectral methods.

“Interaction between concentration of measure: sample size m vs. components k”
“Implications of Theorem 3”

For bounded moments we can follow a similar approach to ‘Anandkumar et al., 2012’ to obtain analogous results using Theorem 2. For unbounded moments, as can be seen from Theorem 3 (the proof is in the appendix), we derived the condition based on n \geq poly(k). In other words, in both cases the inferrable number of components k increases with the sample size m.

Theorem 3 shows that for a given level of confidence sample size m, dimensions and the number of latent attributes are related. We will provide a more intuitive of the dependence in the form of corollaries in the final version of the paper.

“Applicability of spectral methods”

Methods of moments are common tools for statistical inference. For instance, maximum entropy models use this when matching sufficient statistics. In this view our model combines a frequentist approach with a Bayesian statistical model. We prove that when the model is faithful and when we have sufficient data, we will be able to recover the parameters. Whenever these conditions fail, we will, at least, obtain a model that is close in terms of moments. Traditional methods like MCMC and variational inference typically do not come with such guarantees. Our experimental results justify this claim.

“Truncation level of k”

The parameter \epsilon on Line 223 is a threshold determining the truncation level of k. In our paper this is obtained by the largest slope on the curve of sorted eigenvalues. For synthetic data there is no ambiguity since the data was generated using 4 latent features. On real-world data this is often less clear since it becomes a matter of significance of features that are important enough. This is also a reason why in the IBP, the assumption of having potentially infinite of latent features(dishes) is allowed. That said, it is also the sample size that allows us to determine features with sufficient accuracy (see the above theoretical bounds).

“Number of random restarts in the power iteration”

In line 375 we discuss that the 100 dimensional image only needs 10 random restarts. For larger datasets such as the gene expression data 30 random restarts are used. Both this number and the number of iterations depend on the dataset.

“Dirichlet Process \neq Dirichlet Distribution”

We thank the referees for catching this incorrect description. We meant the Dirichlet Distribution rather than the Process. In fact, the latter is an exciting research question since higher order moments as arising from the underlying reference measure G_0 are typically non-trivially related to those of the Dirichlet Process.

We thank the reviewers again for all the suggestion on both theoretical and experimental aspects of our paper. We will fix the notation and spelling mistakes noted by the reviewers.